# Optimal government and manufacturer incentive contracts for green production with asymmetric information

Jiayang Xu[1]*, Jian Cao[2,3], Sanjay Kumar[4], Sisi Wu[1]

1 School of Economics and Management, Zhejiang University of Science and Technology, Hangzhou, Zhejiang Province, China, 2 School of Management, Zhejiang University of Technology, Hangzhou, Zhejiang Province, China, 3 Center for Global & Regional Environmental Research, The University of Iowa, Iowa City, Iowa, United States of America, 4 College of Business, Valparaiso University, Valparaiso, Indiana, United States of America

* jiayangxu@zust.edu.cn

## Abstract

Governments commonly utilize subsidy policy to incentivize manufacturers to produce green products, promoting sustainable development. However, in the presence of information asymmetry, some manufacturers may dishonestly misrepresent the green degree of their products to secure higher subsidies. This study examines different incentive contracts between the government and a green product manufacturer who keeps private information of a product's green-degree in a principal-agent model. Lump-sum transfer and fixed- and flexible-proportion benefit-sharing contracts are proposed to investigate screening and improving green-degree issues. To further enhance the flexible-proportion benefit-sharing contract, we construct a non-linear coordinated contract based on the Nash bargaining solution. The revelation principle and Nash bargaining are performed for comparison and analysis of the contracts. We find that the lump-sum contract reveals true green-degree information but fails to impel manufacturers to improve product's green-degree in developing countries where green product development is in initial stages. In contrast, both fixed- and flexible- proportion benefit-sharing contracts are effective in reveling and enhancing green-degree. The non-linear coordination contract optimizes resource allocation and achieves Pareto improvement. An applied case study for inkjet printer operations and numerical experiments corroborate our model findings.

## 1. Introduction

The industrial revolution and economic progress are sometimes attributed to having caused resource overconsumption and environmental deterioration. Environmental-related issues such as global warming, ozone depletion, water and air pollution, and the resulting extinction of species have become important issues in the developing economies of India, Brazil, and China [1, 2]. Researchers have appealed for production and use of green product to alleviate resources and environmental problems, because green products are characterized by energy saving, low pollution, recyclability and renewability [3–5].

**Data Availability Statement:** All relevant data are within the paper and its Supporting information files.

**Funding:** This study was financially supported by the National Natural Science Foundation of China

(https://www.nsfc.gov.cn/) in the form of grants (71874159 & 72274175) awarded to JC. This study was also financially supported by the State Key Program of National Social Science Fund of China (http://www.nopss.gov.cn/) in the form of a grant (22AZD127) awarded to JC. This study was also financially supported by the Zhejiang Office of Philosophy and Social Science in the form of a grant (22NDJC057YB) awarded to JC. This study was also financially supported by the Humanities and Social Sciences Youth Foundation of Ministry of Education of China (http://www.moe.gov.cn/) in the form of a grant (22YJC630178) awarded to JX. The funders had no role in study design, data collection and analysis, decision to publish, or preparation of the manuscript.

**Competing interests:** The authors have declared that no competing interests exist.

The preference and popularity of green products are relevant to market forces as well as government agencies. Although many producers have recognized the practical significance of green products compared with traditional products, they are not active in production of green product because the development of green products faces barriers such as high cost, lack of technology, and limited consumer perception and demand for green products [6–8]. One of the significant barriers is the higher production costs associated with green products. Eco-friendly materials and technologies often come at a higher price compared to conventional alternatives. Manufacturers may face difficulties in implementing sustainable practices and transitioning to greener alternatives due to the initial investment required [9]. Additionally, manufacturers may also face technical difficulties in integrating new equipment or processes into their existing production systems when implementing green manufacturing processes [10]. Adapting to and optimizing these technologies requires expertise, research, and development, which can be time-consuming and costly. While there is a growing awareness and interest in green products, the demand for such products may still be limited compared to conventional alternatives. Consumers may prioritize factors such as price, convenience, and product performance over environmental considerations [11]. Manufacturers face the challenge of creating green products that meet consumer expectations in terms of cost, quality, and functionality to drive wider adoption. The development of green product not only relies on the power of the market itself, but also needs government guidance. Governments are assuming an increasingly significant role in the quest for sustainability, seeking to minimize the social and environmental impacts caused by production [12]. Therefore, the government agencies have enacted laws that encourage procuring and using green products. For example, the European Union (EU) has released *Energy-using Products* (EuP, 2005/32/EC), requiring that manufacturers should adopt the eco-friendly green design of product [13]. The US Environmental Protection Agency has developed environmentally preferable purchasing guidelines that focus on reducing raw material consumption, producing using renewable materials, and recycling used products [14]. A *Guide of Recycle Development* by the State Council in China mandates the establishment of a government preferential procurement system for remanufactured goods [15]. In general, environmental regulations effectively guide manufacturers towards green production transformation and pollution reduction though directly affecting production decision in firm level, such as resource reallocation, capital investment and innovation incentives.

Green products have recyclable materials and are manufactured using energy-saving technology to reduce waste and the volume of disposed of toxins [16, 17]. We use the term *green-degree* as a measure of the proportion of reused components in the finished product. A high green-degree product is manufactured using a high percentage of environmentally friendly remanufactured parts. Remanufacturing is a process by which previously sold, worn, or non-functional products or components are returned to its original performance with a warranty equivalent to a new product [18, 19]. Typically, the production of a remanufactured product requires the implementation of a set of working procedures such as product disassembly, classification, component testing, restoration, and reassembly. All of these procedures may not be monitored and controlled by government agencies [20, 21].

To encourage green design and production, government agencies in various countries have subsidized the consumption of low-carbon products, such as environment-friendly refrigerators, energy-saving air conditioners, new energy vehicles and so on [22, 23]. However, in policy design and implementation, the government is at a disadvantage because of information asymmetry. The manufacturers have internal information about the green-degree of a product, but they may not reveal such information to the government agencies. A high green-degree product may incur extra costs for manufacturers due to production technology innovations, production equipment improvement, and environmentally friendly raw materials purchase.

To get higher subsidies, manufacturers may have incentives to provide an inflated level of green-degree information when reporting to government agencies. Furthermore, it may be burdensome and expensive for government agencies to audit manufacturers to uncover the true green-degree of a product. For example, in 2016, the ministry of finance of China announced a special inspection of subsidies for the promotion and application of new energy vehicles and publicly exposed five typical cases of illegal cheating of financial subsidy for new energy vehicles. Notably, the largest reported amount of fraudulent subsidies amounted to 520 million CNY (~73 million $). Given the asymmetry of green-degree information, the government faces challenges in determining the appropriate level of subsidies and how they should be allocated. Consequently, finding solutions to prevent fraudulent subsidy claims by manufacturers and ensuring the efficient and equitable utilization of government's green subsidies becomes an immediate priority.

It can be found that an incentive mechanism may effectively avoid the waste of government subsidy expenditure and unreasonable use when the government cannot clearly know the green-degree information of enterprises. On the premise that the green-degree of products can be improved, this paper uses the principal-agent theory to study the design of incentive contracts between government agencies and manufacturers who have private information about their product's green-degree. In this mechanism, the government is the principal and the manufacturer is the agent. The government first designs a set of contract menus. After the manufacturer observes the contract, it chooses according to the contract requirements [24–26]. A lump-sum transfer contract and two linear benefit-sharing contracts are proposed, where the order volume and transfer payment for green products are treated as two primary decision variables for the government agency. In the lump-sum contract, the government offer a fixed transfer payment to the manufacturer. The linear benefit-sharing contract is divided into fixed- and flexible-proportion contracts. A non-linear coordination contract, using the Nash bargaining model, is proposed as an improvement over the flexible-proportion contract. The non-linear contract achieves Pareto improvement for the government and the manufacturer as well as social welfare enhancement. We further explore numerical experiments to understand the relationship between various parameters. A practical example using an inkjet printer is presented to show the implications and usefulness of contracts developed in this paper.

This paper helps guide the governments in developing countries to design incentive mechanisms on green products and encourages firms to produce environment-friendly products for socially sustainable development. The remainder of this paper is organized as follows. Section 2 discusses the related literature. In Section 3, we describe the model. Section 4 designs the lump-sum transfer contract under asymmetric information. Two linear benefit-sharing contracts in fixed- and flexible-proportion benefit-sharing settings are proposed in Section 5. A non-linear coordination contract based on Nash bargaining is provided in Section 6. Section 7 presents numerical analysis, and Section 8 illustrates application to inkjet printer manufacturing while concluding remarks and managerial implications are presented in Section 9.

## 2. Literature review

The literature related to our study can be categorized into three streams, incentive contracts under asymmetric information, game analysis on green product improvement, and incentives of government intervention.

### 2.1. Incentive contracts under asymmetric information

A part of literature related to asymmetric information in supply chains explores incentive strategies to reveal or share private information. Previous literature has extensively examined

information asymmetry in relation to different types of information, including demand information [27–29], quality information [30, 31], cost information [32, 33] and price information asymmetry [34, 35]. In numerous instances, the presence of information asymmetry can affect the decision-making of participants in the game and imposes additional burdens on them. Samuelson [36] investigated a two-person bargaining game, which reflects that consideration of asymmetric information precludes the attainment of mutually beneficial agreements. Pavlov et al. [37] showed how incomplete information regarding preferences for fairness affects the performance of a supply chain. Diverging from conventional research on information asymmetry, this paper primarily concentrates on the information asymmetry concerning manufacturers' green-degree and considers government transfer payment as a main decision -making variable.

To address the problems caused by asymmetric information, many scholars explored appropriate contracts in supply chain. With asymmetric information about demand distribution, Babich et al. [38] derived a menu of optimal (for the supplier) buyback contracts where the supplier must give up some profit in return for being less informed than the retailer. In the study of Çakanyıldırım et al. [39], the retailer shares a part of channel profit with the supplier who has the private information of its unit production cost to maximize expected profit. Giovanni [40] constructed two profit-sharing contracts to explore how the sharing fraction should be determined to make both manufacturer and retailer economically better off in a closed-loop supply chain. Hosseini-Motlagh et al. [41] proposed a bi-level wholesale contract in a three-tier supply chain. They developed a profit-sharing plan based on the Nash-bargaining model to share the extra profits among supply chain partners. Jung and Kouvelis [42] built a game-theoretic model to capture the firms' strategic interactions under the competitive supply partnership with information leakage. Ghosh and Shah [43] studied two models of cost-sharing and presented the effect of contracts on product green-degree. Hong and Guo [44] established a green-marketing cost-sharing contract where the manufacturer shares a proportion of green-marketing costs with the retailer, which could achieve environmental improvements. Toktaş-Palut [45] developed an integrated two-part tariff contract to explore how to increase the remanufacturer's greening level. Most papers mentioned above primarily focus on a single means for profit or revenue sharing; that is, the principal shares a fixed fraction of its profit with the agent in the menus of a contract. In addition to a fixed proportion in profit sharing, in this paper, we propose a flexible proportional profit-sharing contract in which the shared percentage is related to the green-degree of the products.

## 2.2. Game analysis on green product

Green product development has become an important issue due to the direct impact of green products on the environment. Achieving green products requires a comprehensive examination of environmental issues throughout the entire life cycle, from design to the disposal phase of old products [46–48]. Thus, the evaluation of the green degree of products is crucial. Xu et al. [49] provided a quantitative evaluation method to measure the green degree of different products of the same use function with an indicator system established, including fundamental indicators, general indicators, and leading indicators. Wang et al. [50] introduced a comprehensive method that integrates Fuzzy Extent Analysis and Fuzzy TOPSIS for the assessment of environmental performance with respect to different product designs. As a kind of increasing popular green products, the manufactured products' sustainability performance evaluation have been extensively studied. Golinska et al. [51] classified remanufacturing sustainability performance by taking energy consumption level, waste generation level, material recovery rate and generated emissions level as criteria. Xu [52] developed an assessment model of

resource and environmental benefit for the remanufacturing of decommissioned construction machinery to analyze its energy, materials and carbon dioxide emissions.

Most of the studies on specific decision-making processes of green products are based on the framework of game theory. Fairchild [53] examined the stimulus for manufacturing companies to produce green product from a game-theoretic perspective. Amato et al. [54] modeled a game-theoretic framework to expound competitive advantage derived from green practices and remanufactured products, noting that increasing demand for eco-friendly products drives firms to develop dedicated environmental manufacturing and marketing strategies. Jamali and Rasti-Barzoki [55] illustrated the influence of different decision parameters on the greenness of a product, exploring the pricing of green and non-green products using a game theory approach. Kannan et al. [56] proposed a fuzzy criteria approach to help manufactures to select the best supplier and demonstrated the effectiveness of the proposed approach. Zhang et al. [57] focused on the green supply chain performance in a single manufacturer-retailer setup to analyze equilibrium strategies of open-loop, feedback and myopic games in decentralized supply chain. Hong et al. [58] studied the competitive pricing of green products and non-green products under a Nash game framework, discussing the relationship between pricing strategy, product quality, and consumers' environmental awareness.

In the above-mentioned research, scholars have employed traditional game theory, such as Cournot and Stackelberg games, in the context of the green supply chain. This study expands the previous works of game modelling by introducing incentive contracts based on principal-agent theory. Instead of monitoring and evaluating the green-degree of products, we utilize the principal-agent model to screen the asymmetric green-degree information. Our work systematically analyzes different types of screening, efficient incentives, and profit distribution in green product development based on principal-agent model. By considering the profits of the principal and the agent, we aim to identify the most favorable mechanism which could motivate manufacturers to improve product green-degree.

## 2.3. Incentives of government intervention

The government plays a very important role in promoting manufacturers' green production. There is a body of literature focused on government policy or decision-making regarding manufacturers' green production. Numerous articles have investigated the impact of government subsidies or tax policies on the production and business activities of enterprises. Lombardini-Riipinen [59] studied the optimal quality subsidy (emission tax) and uniform ad valorem tax policies to improve environmental quality. Mitra and Webster [60] examined the effects of government subsidies as a means to promote remanufacturing activity by considering a subsidy proportional to remanufacturing volume. Yi and Li [61] examined the manufacturer-retailer channel and discovered that government subsidies for energy efficiency resulted in a decrease in carbon emissions, while carbon taxes did not consistently have the same effect. Wu et al. [62] used the ordinary least square method to assess the influence of government research and development subsidies on enterprises' investments in renewable energy, as well as the role of ownership attribute variations in shaping this impact. Mondal and Giri [63] explored how government subsidies effectively expanded sales volume by enhancing the greening level of products.

The aforementioned research analyzes how government policies, as external factors, affect the producers' production decision and profit, while government green procurement has also been discussed. Palmujoki et al. [64] proposed and examined environmental criteria in public procurement contracts. Simcoe and Toffel [65] argued that government procurement policies could lead to spillover effects that elevate the environmental standard of products and affect

private-sector demand for similar products. Sparrevik et al. [66] considered the procurement of an environmentally friendly office building and proposed measures to improve the traditional green public procurement process. Focusing on green production and sustainable products, Hafezalkotob and Zamani [67] investigated decision-making for suppliers who provide raw material, as well as manufacturers engaged in producing products with different levels of environmental pollution under government financial supervision to reduce environmental pollution.

Few studies have applied incentive contracts between the government and enterprises to address the issue of subsidies and green production strategies under asymmetry information. In the context of government subsidies and green procurement, this paper investigates the transfer payment from government and the green production strategy of manufacturer through the principal-agent contract. The goal is to seek the optimal strategy for both players, while considering the manufacturer's private information regarding its green- degree.

### 2.4. Contributions to literature

This paper makes contributions to the literature in the following four aspects.

- There is existing literature on incentive measures in revenue-sharing contracts in a supply chain under symmetric information, as well as supply chain coordination and collaboration under asymmetric information. Our research contributes to the literature by designing an incentive contract between the authority (e.g., government) and supply chain members (e.g., manufacturers) in the context of asymmetric information.

- Prior studies have focused on the government's financial intervention on green products but have neglected its role in contract design. We develop contracts between the government and manufacturers where the government prioritizes the procurement of green products. This approach allows the government to exert influence on the market for green products.

- In addition to a lump-sum transfer payment that some previous studies have investigated, our research introduces two benefit-sharing contracts between the government and a manufacturer. Benefit-sharing contracts are commonly used in supply chain coordination to optimize the interests of all parties involved. Our research is novel in integrating green-degree financial incentives into contracts. We also conduct an analysis to validate the effectiveness of the proposed contracts.

- Our paper is geared towards developing countries where green product development is in initial stages. Our intention is to provide insights that can assist government agencies in formulating policies that encourage manufacturers to disclose and enhance the greenness of their products in countries such as China, India, and Brazil.

More specifically, our study focuses on contracts between a risk-neutral government and a risk-neutral green product manufacturer that privately knows its product green-degree. For a product with green-degree $g$, the government determines the procurement volume of green products $q(g)$ and transfer payment $t(g)$ in the incentive contract, which aids in revealing the manufacturer's true green-degree. For notations, see S1 Table. Three contracts, namely, lump-sum transfer payment, linear fixed-proportional benefit-sharing, and linear flexible-proportional benefit-sharing are compared for their incentives on the green-degree improvement. To further enhance the flexible-proportion benefit-sharing contract which is effective in screening and improving green-degree, we construct a non-linear coordinated contract based on the Nash bargaining solution.

## 3. The model

In this section, we develop a game model between the manufacturer (agent) and the government (principal) dealing with the procurement of green products. Demand parameters for green products, production cost, transfer payment, manufacturer's profit, and government's benefit are also identified.

### 3.1. The product

We model a scenario when product greenness or green-degree $g$ is privately known to the manufacturer (agent) but is unobservable by the government (principal). $g$ is continuously distributed on the support $G = [\underline{g}, \bar{g}]$. In particular, we assume $g \geq 0$. The uncertainty about the green-degree is represented by a probability distribution $F(g)$ with associated density function $f(g)$ that is strictly positive, with $f'(g) \leq 0$ in the early stages of green product development. The government's demand for green products is related to the procurement price $p$ and green-degree $g$, i.e., $q = q(p, g)$. This demand function follows $q_p < 0$, $q_{pp} \geq 0$, $q_g > 0$, and $q_{gg} \geq 0$. The government's demand function $q = a - bp + hg^2$ has a negative linear correlation with $p$ and is proportional to $g^2$. Here, $a$ is the base market capacity of products, $b$ is the price elasticity coefficient of demand, and $h$ is the attraction coefficient of $g^2$, indicating consumers' preference for green products; $a, b, h > 0$. Then, the price function can also be expressed as $p = a_1 - a_2q + a_3g^2$, where $a_1 = a/b$, $a_2 = 1/b$, $a_3 = h/b$, with $a_1, a_2, a_3 > 0$.

### 3.2. The manufacturer

To produce green products, the manufacturer may incur extra green costs related to investment in R&D, specialized production equipment, and procurement of environment-friendly raw material. These costs may be high in developing economies such as China, where the green market is still at its initial stages. We define per unit production cost for green products $c_g = \underline{c} + 0.5zg^2$. Transfer payment from the government to a risk-neutral manufacturer is $t(g)$. Therefore, the manufacturer's profit is represented by

$$\pi_M = (p - c_g)q(g) + t(g) = [a_1 - a_2q(g) + (a_3 - 0.5z)g^2 - \underline{c}]q(g) + t(g) \tag{1}$$

where $\underline{c}$ is the unit cost of conventional products, $z$ is deemed as an extra green cost factor, $0.5zg^2$ signifies the additional cost per unit of green products, and $q(g)$ is the government's demand for remanufactured products.

### 3.3. The government

The government purchases green products preferentially. The *environmental benefit* is based on a product's green-degree and volume. For the model in this paper, this environmental benefit is treated similarly to economically desired benefit by the government. Considering that the governments may have social concerns in addition to environmental concerns, they also care about the revenue of the manufacturer. Additionally, depending on the actual price paid, the government could obtain *Consumer Surplus* (CS), which is included in the utility function. See S1 Fig for additional details of CS. When the government purchases green products from a manufacturer, an additional monetary transfer $t(g)$ is provided to the manufacturer. The government is risk-neutral; thus, its utility function, which includes the utility derived from

environmental benefit for green products, can be expressed as:

$$U_G = vgq(g) - pq(g) + (p - c_g)q(g) + a_2 q(g)^2/2 - t(g)$$
$$= \left( vg - \underline{c} - 0.5zg^2 \right) q(g) + a_2 q(g)^2/2 - t(g)$$

(2)

where $v$ is the coefficient of environmental benefit per unit of a green product and may depend on the product's remanufacturability and the extent of environmental pollution it causes. Therefore, $v$ varies with distinct types of products and the production process. For instance, $v$ value for home appliances could be higher than that for furniture duo to differences in the raw materials used and the difficulty in the post-use environmental impact of the product. $vgq(g)$ represents the total environmental benefit generated by green products. The government's utility function or function of environmental benefit is similar to that discussed in the works of Bovenberg and Goulder [68] and Walls and Palmer [20]. Note that often the cost of waste treatment is covered by taxes the government levies from taxpayers. When compared to traditional products, high green-degree products may need lower expenses in end-of-life treatment of products. $vgq(g)$ could be regarded as the reduction in disposal and treatment costs because of the green-degree of products. In addition to the purchase cost $pq(g)$, the government considers the manufacturer's revenue $(p - c_g)q(g)$. $a_2 q(g)^2/2$ is the consumer surplus. See S1 Fig.

### 3.4. Model framework

This paper is primarily focused on developing regions like India and China, where green product development is still at its embryonic stage. Consumers are relatively unaware of green products in this early period, and manufacturers may need government policy intervention to encourage them to produce green products. Considering these factors in developing economies, the attraction coefficient of greenness $h$ is expected to be lesser than the price elasticity coefficient $b$. Also, the additional cost factor $z$, associated with producing green products, is expected to be relatively high.

Under the asymmetric information conditions, the green-degree $g$ of products is privately held by manufacturers and is unobservable to the government. The government (principal) offers a contract that provides incentives for the manufacturer (agent) to reveal and stimulate their product green-degree $g$. The manufacturer's objective is to maximize its profit, while the government aims to maximize its utility from green products. Both the principal and the agent are risk-neutral, with their expected utility equivalent to the expected profits.

Focusing on the positive effects of green products on the resources utilization and environmental protection, we regard the green products' environmental benefit as the economic revenue to the government. Note that green products may help reduce disposal and treatment costs covered by the government. For remanufactured products, greenness is reflected in remanufacturing progress wherein a higher green-degree is attributed to products that contain more spare parts from recycling. The green-degree of conventional products without any recycling part is $g = 0$, while products entirely made from secondary materials or components have a maximum value $g = 1$. Besides, the government considers the revenue of manufacturer and can obtain another benefit from procurement, measured by *consumer surplus*. *Consumer surplus* is affected by the order quantity of green products and the difference between the maximum price that the government, as a consumer, is willing to pay and the actual price paid.

Our focus in this paper is on remanufactured green products. According to the Chinese Government's *Guide of Recycle Development* [15], the government is required to preferentially procure remanufactured products that are conductive to environmental protection and

resource utilization. For instance, Gasoline-powered auto manufacturers can produce new auto components with raw materials or by remanufacturing used parts that can potentially reduce pollution as well as save 60% of energy and 70% of materials in the production process [69].

## 4. Lump-sum transfer contract

We first consider a contract where the government offers a lump-sum transfer payment $t(g)$ to incentivize manufacturers to reveal their green-degrees. The environmental benefit of a green product with volume $q(g)$ is $vgq(g)$. The manufacturer's revenue from selling products to the government is $(p - c_g)q(g)$, which is also a part of the government's benefit. $pq(g)$ is the procurement cost, while $CS$ is $a_2q^2(g)/2$. Therefore, the government's utility function is $U_G = (vg - \underline{c} - 0.5zg^2)q(g) + a_2q(g)^2/2 - t(g)$, and its expected profit is $EU_G = \int_{\underline{g}}^{\bar{g}} U_G f(g)dg$. A manufacturer can earn revenue $[a_1 - a_2q(g) + a_3g^2]q(g)$ and incurs a cost $(\underline{c} + 0.5zg^2)q(g)$, such that its profit function is $\pi_M = [a_1 - a_2q(g) + (a_3 - 0.5z)g^2 - \underline{c}]q(g) + t(g)$. The process of contract design can be expressed as the following programming problem.

$$(P1): \max_{\{t(g), q(g)\}} EU_G = \int_{\underline{g}}^{\bar{g}} [(vg - \underline{c} - 0.5zg^2)q(g) + a_2q^2(g)/2 - t(g)]f(g)dg$$

$$s.t \arg\max_{\{g\}} \pi_M = \arg\max_{\{g\}} \{[a_1 - a_2q(g) + (a_3 - 0.5z)g^2 - \underline{c}]q(g) + t(g)\} \tag{3}$$

$$\pi_M = [a_1 - a_2q(g) + (a_3 - 0.5z)g^2 - \underline{c}]q(g) + t(g) \geq \underline{\pi}_M \tag{4}$$

The *incentive compatibility* (IC) constraint in Eq (3) states that the manufacturer could maximize its profit by choosing a contract consistent with its true green-degree of the product. Eq (3) ensures that $g$-type manufacturer reports its green-degree $g$ truthfully (i.e., does not report a false level of greenness $\tilde{g}$, $\tilde{g} \neq g$). The *individual rationality* (IR) constraint in Eq (4) guarantees that the manufacturer earns at least its reservation profit when choosing the contract designed for its level of green-degree. $\underline{\pi}_M$ represents the reservation profit, which is the profit the manufacturer would make by adopting other contracts or producing conventional (non-green) products while rejecting this commission. The government orders may prioritize green products, and if the supply of green products cannot meet the demand, some conventional products may be procured.

The *revelation principle* [70, 71] is used to solve the problem (P1). For proofs, see S1 Appendix. Let the profit of type $\underline{g}$ manufacturer be $\underline{\pi}_M$. Problem (P1) can be transformed into (P1'),

$$(P1'): \max_{\{\pi_M(g), q(g)\}} EU_G = \int_{\underline{g}}^{\bar{g}} \{[vg + a_1 - 0.5a_2q(g) + (a_3 - z)g^2 - 2\underline{c}]q(g) - \pi_M\}f(g)dg$$

$$s.t. \dot{\pi}_M(g) = 2(a_3 - 0.5z)gq(g) \tag{5}$$

$$\pi_M(g) \geq \underline{\pi}_M \tag{6}$$

where Eqs (5) and (6) are manufacturer's IC and IR constraints, respectively. When $a_3 - 0.5z \geq 0$, we have $\dot{\pi}_M(g) \geq 0$.

Inserting $\underline{\pi}_M$ into Eq (6) yields

$$\pi_M(g) = \int_{\underline{g}}^{\bar{g}} 2(a_3 - 0.5z)\tau q(\tau)d\tau + \underline{\pi}_M \tag{7}$$

Thus

$$
\begin{aligned}
\int_{\underline{g}}^{\bar{g}} \pi_M(g)f(g)dg &= \underline{\pi}_M + 2\int_{\underline{g}}^{\bar{g}} \left\{ \int_{\underline{g}}^{g}(a_3 - 0.5z)\tau q(\tau)d\tau \right\}f(g)dg \\
&= \underline{\pi}_M + 2\int_{\underline{g}}^{\bar{g}} \left[ \int_{g}^{\bar{g}} f(\tau)d\tau \right](a_3 - 0.5z)gq(g)dg \\
&= \underline{\pi}_M + 2\int_{\underline{g}}^{\bar{g}} \frac{1 - F(g)}{f(g)}(a_3 - 0.5z)gq(g)f(g)dg
\end{aligned}
\tag{8}
$$

Plugging Eq (8) into the objective function in (P1′), we express the government's expected utility as:

$$EU_G = \int_{\underline{g}}^{\bar{g}} \left\{ vg + a_1 - 0.5a_2q(g) + (a_3 - z)g^2 - 2\underline{c} - \frac{2[1 - F(g)]}{f(g)}(a_3 - 0.5z)g \right\}q(g)f(g)dg$$
$$- \underline{\pi}_M \tag{9}$$

By taking derivative of the objective function with respect to $q(g)$, we obtain second-best volume,

$$q_{(1)}^{SB}(g) = \frac{1}{a_2}[a_1 - 2\underline{c} + vg + (a_3 - z)g^2] - \frac{2(a_3 - 0.5z)g[1 - F(g)]}{a_2 f(g)} \tag{10}$$

where superscript *SB* refers to the second-best solution under asymmetric information, and subscript (1) denotes (P1). We have $\partial q_{(1)}^{SB}(g)/\partial v > 0$, which indicates that purchase volume is positively correlated with environmental benefit coefficient *v*.

Based on Eq (10) we calculate the first-order derivative of $q_{(1)}^{SB}(g)$ and state the following proposition.

**Proposition 1**. Given the lump-sum transfer incentive contract $\left\{ t_{(1)}^{SB}(g), q_{(1)}^{SB}(g) \right\}$, the monotonicity of $q_{(1)}^{SB}(g)$ in g depends on $a_3$ and z:

1. $\partial q_{(1)}^{SB}(g)/\partial g \geq 0$, if $a_3 \geq 0.5z$;

2. $\partial q_{(1)}^{SB}(g)/\partial g < 0$, if $a_3 < 0.5z$.

Proposition 1 provides corresponding ordering strategies for the government. In a market where consumers have a preference for green products and are willing to pay a premium for environmentally friendly products, and if the green production cost is not high ($a_3 \geq 0.5z$), the government may purchase a larger quantity of products with higher green-degree. However, in developing economies where green product development is still in its early stages ($a_3 < 0.5z$), the government procurement volume for green products decreases with an increase in the green-degree of products.

From Eqs (4) and (7), we further obtain

$$t_{(1)}^{SB}(g) = 2\int_{\underline{g}}^{g} (a_3 - 0.5z)\tau q_{(1)}^{SB}(\tau)d\tau - \left[a_1 - \underline{c} + (a_3 - 0.5z)g^2 - a_2 q_{(1)}^{SB}(g)\right]q_{(1)}^{SB}(g) + \underline{\pi}_M \quad (11)$$

$$\pi_{M,(1)}^{SB}(g) = \underline{\pi}_M + 2\int_{\underline{g}}^{g} (a_3 - 0.5z)\tau q_{(1)}^{SB}(\tau)d\tau \quad (12)$$

$$U_{G,(1)}^{SB}(g) = (vg - \underline{c} - 0.5zg^2)q_{(1)}^{SB}(g) + 0.5a_2\left[q_{(1)}^{SB}(g)\right]^2 - t_{(1)}^{SB}(g) \quad (13)$$

**Proposition 2**. Given the lump-sum transfer incentive contract $\left\{t_{(1)}^{SB}(g), q_{(1)}^{SB}(g)\right\}$,

1. $\dot{\pi}_{M,(1)}^{SB}(g) \geq 0$, if $a_3 \geq 0.5z$;

2. $\dot{\pi}_{M,(1)}^{SB}(g) < 0$, if $a_3 < 0.5z$.

Since $g \geq 0$ and $q_{(1)}^{SB}(g) \geq 0$, from Eq (5) we can deduce that $\pi_{M,(1)}^{SB}(g)$ is an increasing function of $g$ when $a_3 \geq 0.5z$. This implies that the monotone hazard ratio allows the government to screen manufacturers with different green degrees. $\dot{\pi}_{M,(1)}^{SB}(g) \geq 0$ signifies that manufacturers can be incentivized to improve green-degree using this contract. On the contrary, if $a_3 < 0.5z$, $\pi_{M,(1)}^{SB}(g)$ is a decreasing function of $g$. This indicates that the contract can still differentiate between manufacturers with different green degrees, but it would fail to provide incentives for green degree improvement. Hence, the lump-sum contract would only be effective in markets where customers are already aware of green products, i.e., $a_3 \geq 0.5z$.

Note that $a_3$ represents the influence of green-degree on price, and $z$ is intended to be the extra cost factor for green production. If $a_3 < 0.5z$, according to Propositions 1 and 2, both the order quantity and manufacturer's profit decrease with an increase in green-degree. However, when $a_3 \geq 0.5z$, both the government's order quantity and the manufacturer's profit increase with the product's greenness. In practice, green products are still in the early stages of development in developing economies, consumer awareness towards such products is inadequate, and producers may incur costly investments in green technologies. Accordingly, in these economies, $a_3 \geq 0.5z$ may not hold true. This condition may make the lump-sum contract difficult or infeasible to implement. In the next section, we propose two benefit-sharing contracts.

## 5. Benefit-sharing contract

We propose two linear benefit-sharing contracts from a cooperative perspective. In addition to the fixed payment $t(g)$, a portion of the environmental benefits derived from green products is transferred to the manufacturer through financial compensation. This serves as an incentive for the production of higher green-degree products. We consider fixed and flexible benefit-sharing, which are analyzed in linear fixed- and flexible-proportional benefit-sharing contracts, respectively.

### 5.1. Benefit-sharing contract in linear fixed-proportion

Green products have the potential to reduce the government's expenses on environmental pollution management. As a result, the government may choose to share a portion $\alpha$ of the benefit $vgq(g)$ derived from green products with the manufacturer. The government retains the remaining share $1-\alpha$, $\alpha \in [0,1]$. In this contract, in addition to the transfer payment $t(g)$ considered in the lump-sum contract, the manufacturer also receives additional $\alpha vgq(g)$ owing to

its green production. Therefore, a linear fixed-proportional benefit-sharing contract is expressed as

$$(P2): \max_{\{t(g),q(g)\}} EU_G = \int_{\underline{g}}^{\bar{g}} \{[v(1-\alpha)g - \underline{c} - 0.5zg^2]q(g) + a_2q^2(g)/2 - t(g)\}f(g)dg$$

$$s.t. \arg\max_{\{g\}} \pi_M = \arg\max_{\{g\}} \{[a_1 - a_2q(g) + (a_3 - 0.5z)g^2 - \underline{c} + \alpha vg]q(g) + t(g)\} \quad (14)$$

$$\pi_M = [a_1 - a_2q(g) + (a_3 - 0.5z)g^2 - \underline{c} + \alpha vg]q(g) + t(g) \geq \underline{\pi}_M \quad (15)$$

The manufacturer's IC and IR constraints are expressed by Eqs (14) and (15), respectively. The *revelation principle* is used to solve the problem (P2). Using an analysis similar to that in Section 4, problem (P2) can be transformed to (P2′),

$$(P2'): \max_{\{\pi_{M(g)},q(g)\}} EU_G = \int_{\underline{g}}^{\bar{g}} \{[vg + a_1 + (a_3 - z)g^2 - 2\underline{c}]q(g) - a_2q(g)^2/2 - \pi_M\}f(g)dg$$

$$s.t. \dot{\pi}_M(g) = [2(a_3 - 0.5z)g + \alpha v]q(g) \quad (16)$$

$$\pi_M(g) \geq \underline{\pi}_M \quad (17)$$

For $\alpha \geq (z - 2a_3)\bar{g}/v$, we have $\dot{\pi}_M(g) \geq 0$. The condition $z - 2a_3 \geq 0$ can be satisfied in the early development stages of green products, so $\alpha > 0$.

Simplifying $\pi_M(\underline{g})$ for $\underline{\pi}_M$, and inserting $\underline{\pi}_M$ into Eq (16) yields

$$\pi_M(g) = \int_{\underline{g}}^{g} [2(a_3 - 0.5z)\tau + \alpha v]q(\tau)d\tau + \underline{\pi}_M \quad (18)$$

Thus

$$\int_{\underline{g}}^{\bar{g}} \pi_M(g)f(g)dg = \underline{\pi}_M + \int_{\underline{g}}^{\bar{g}} \frac{1 - F(g)}{f(g)}[2(a_3 - 0.5z)g + \alpha v]q(g)f(g)dg \quad (19)$$

Substituting Eq (19) into the objective function in (P2′), the government's expected utility function can be expressed as:

$$EU_G = \int_{\underline{g}}^{\bar{g}} \left\{ vg + a_1 - 0.5a_2q(g) + (a_3 - z)g^2 - 2\underline{c} - \frac{1 - F(g)}{f(g)}[2(a_3 - 0.5z)g + \alpha v] \right\}q(g)f(g)dg$$

$$- \underline{\pi}_M \quad (20)$$

We get the first-order condition $\partial EU_G/\partial q(g) = 0$ and second-best volume as:

$$q_{(2)}^{SB}(g) = \frac{1}{a_2}[a_1 - 2\underline{c} + vg + (a_3 - z)g^2] - \frac{[2(a_3 - 0.5z)g + \alpha v][1 - F(g)]}{a_2f(g)} \quad (21)$$

where subscript (2) denotes formulation (P2). From Eq (21), we have the following proposition.

**Proposition 3**. Under $\alpha \geq (z - 2a_3)\bar{g}/v$, $\partial q_{(2)}^{SB}(g)/\partial g > 0$ and $\partial q_{(2)}^{SB}(g)/\partial \alpha < 0$.

With the premise of $\alpha \geq (z - 2a_3)\bar{g}/v$, the volume of green procurement by the government is monotonically associated with the green-degree of the manufacturer. That is, products

with different green-degrees lead to different quantities $q_{(2)}^{SB}(g)$. A Higher green-degree encourages a higher procurement quantity from the government. Conversely, a negative correlation between $q_{(2)}^{SB}(g)$ and $\alpha$ signifies that the order volume of green products reduces with an increase in the benefit-sharing ratio to the manufacturer.

According to Eqs (15) and (18), we further obtain

$$t_{(2)}^{SB}(g) = \int_{\underline{g}}^{g} [2(a_3 - 0.5z)\tau + \alpha v] q_{(2)}^{SB}(\tau) d\tau$$
$$- \left[ a_1 - \underline{c} + \alpha v g + (a_3 - 0.5z)g^2 - a_2 q_{(2)}^{SB}(g) \right] q_{(2)}^{SB}(g) + \underline{\pi}_M \qquad (22)$$

$$\pi_{M,(2)}^{SB}(g) = \underline{\pi}_M + \int_{\underline{g}}^{g} [2(a_3 - 0.5z)\tau + \alpha v] q_{(2)}^{SB}(\tau) d\tau \qquad (23)$$

$$U_{G,(2)}^{SB}(g) = [v(1-\alpha)g - \underline{c} - 0.5zg^2] q_{(2)}^{SB}(g) + a_2 \left[ q_{(2)}^{SB}(g) \right]^2 \Big/ 2 - t_{(2)}^{SB}(g) \qquad (24)$$

**Proposition 4.** Given linear fixed-proportional benefit-sharing contract $\left\{ t_{(2)}^{SB}(g), q_{(2)}^{SB}(g) \right\}$,

1. if $\alpha \geq (z - 2a_3)\bar{g}/v$, then $\dot{\pi}_{M,(2)}^{SB}(g) \geq 0$;

2. if $\alpha < (z - 2a_3)\bar{g}/v$, then $\dot{\pi}_{M,(2)}^{SB}(g) < 0$.

Proposition 4 implies that when $\alpha \geq (z - 2a_3)\bar{g}/v$, manufacturers of different green-degrees satisfy the monotone hazard ratio and the government can screen different types of manufacturers and incentivize them accordingly to encourage them to improve their product green-degrees. If $\alpha < (z - 2a_3)\bar{g}/v$, $\pi_{M,(2)}^{SB}(g)$ is a decreasing function of g, and manufacturers are less inclined to improve the green-degree of their products. Eq (18) shows, when $\alpha \geq (z - 2a_3)\bar{g}/v$, all manufacturers except those of type $\underline{g}$ can obtain strictly positive information rent $\int_{\underline{g}}^{g} [2(a_3 - 0.5z)\tau + \alpha v] q_{(2)}^{SB}(\tau) d\tau$. The higher the green-degree is, the more information rent manufacturers can get, and their profits increase. Note that $\alpha \in [0,1]$ and $(z - 2a_3)\bar{g}/v \leq 1$ are preconditions for the second-best contract $\left\{ t_{(2)}^{SB}(g), q_{(2)}^{SB}(g) \right\}$. $\alpha \in [(z - 2a_3)\bar{g}/v, 1]$, which indicates that the manufacturer of type g should get a benefit-sharing ratio of at least $(z - 2a_3)\bar{g}/v$.

In practice, the fixed benefit-sharing ratio depends on the manufacturer's bargaining power associated with enterprise-scale, brand reputation, and market competitiveness. Therefore, when $\alpha \in [(z - 2a_3)\bar{g}/v, 1]$, the linear fixed-proportional benefit-sharing contract $\{t_{(2)}^{SB}(g), q_{(2)}^{SB}(g)\}$ can distinguish different types of manufacturers and ensures manufacturers of higher green-degree obtain higher profit. This contract overcomes the problem of adverse selection under asymmetric information of product green-degree and realizes highly efficient incentives to the manufacturers.

The linear fixed-proportional benefit-sharing contract of this section requires the precondition $\alpha \geq (z - 2a_3)\bar{g}/v$. However, this condition may not be satisfied in markets with high green-degree. A high-level g may result in high $\alpha$ from the condition $\alpha \geq (z - 2a_3)\bar{g}/v$. Note that in this scenario, the required $\alpha$ may be equal to or greater than 1, resulting in manufacturers gaining 100% ($\alpha = 1$) of the financial proportion benefits offered under the contract. Therefore, the government may suffer from financial burden.

## 5.2. Benefit-sharing contract in linear flexible-proportion

We propose a linear flexible-proportional benefit-sharing contract to reduce the threshold value of the sharing-benefit coefficient and improve incentive efficiency. Let the benefit-sharing proportion be $\alpha = \alpha(g)$ with a positive first-order derivative $\alpha'(g)$, i.e., higher product green-degree manufacturers receive a higher percentage of the subsidy. We assume that $\alpha(g) = \mu g$; herein, $\mu$ denotes the coefficient of flexible-proportional benefit-sharing and $\mu \in [0, 1/\bar{g}]$. Also $\mu g \leq 1$, therefore $\mu \leq 1/\bar{g}$. The revenue shares for the government and manufacturers are $(1 - \mu g)vgq(g)$ and $\mu vg^2 q(g)$, respectively. The resulting incentive mechanism process based on linear flexible-proportional benefit-sharing contract is represented by

$$(\text{P3}): \max_{\{t(g),q(g)\}} EU_G = \int_{\underline{g}}^{\bar{g}} \{[v(1 - \mu g)g - \underline{c} - 0.5zg^2]q(g) + a_2 q^2(g)/2 - t(g)\}f(g)dg$$

$$s.t \ \arg\max_{\{g\}} \pi_M = \arg\max_{\{g\}} \{[a_1 - a_2 q(g) + (a_3 - 0.5z + \mu v)g^2 - \underline{c}]q(g) + t(g)\} \tag{25}$$

$$\pi_M = [a_1 - a_2 q(g) + (a_3 - 0.5z + \mu v)g^2 - \underline{c}]q(g) + t(g) \geq \underline{\pi}_M \tag{26}$$

The *revelation principle* is used to solve (P3). The analysis process is similar to that in Section 4, and we have:

$$q_{(3)}^{SB}(g) = \frac{1}{a_2}[a_1 - 2\underline{c} + vg + (a_3 - z)g^2] - \frac{2(a_3 - 0.5z + \mu v)g[1 - F(g)]}{a_2 f(g)} \tag{27}$$

where subscript (3) signifies the solution of the programming problem (P3). According to Eq (27), the following proposition can be deduced.

**Proposition 5**. *If $\mu \geq (0.5z - a_3)/v$, then $\partial q_{(3)}^{SB}(g)/\partial g > 0$ and $\partial q_{(3)}^{SB}(g)/\partial \mu < 0$.*

It can be seen from Proposition 5 that in the flexible-proportional benefit-sharing contract, when $\mu \geq (0.5z - a_3)/v$, the volume of green products will increase monotonically with green-degree $g$. The government's demand is higher for products with higher green degrees. Also, for each green-degree product, there exists a different level of demand $q_{(3)}^{SB}(g)$, which implies that products with varying green degrees are not pooled together in the government's demand. The procurement volume of green products, $q_{(3)}^{SB}(g)$, is a decreasing function of $\mu$, indicating that the government's purchase volume goes down when the benefit-sharing ratio to the manufacturers rises.

The government transfer payment, as well as the benefits of the manufacturer and the government, are presented as follows,

$$t_{(3)}^{SB}(g) = \underline{\pi}_M + 2\int_{\underline{g}}^{g}(a_3 - 0.5z + \mu v)\tau q_{(3)}^{SB}(\tau)d\tau$$

$$- \left[a_1 - \underline{c} + (a_3 - 0.5z + \mu v)g^2 - a_2 q_{(3)}^{SB}(g)\right]q_{(3)}^{SB}(g) \tag{28}$$

$$\pi_{M,(3)}^{SB}(g) = \underline{\pi}_M + 2\int_{\underline{g}}^{g}(a_3 - 0.5z + \mu v)\tau q_{(3)}^{SB}(\tau)d\tau \tag{29}$$

$$U_{G,(3)}^{SB}(g) = [v(1 - \mu g)g - \underline{c} - 0.5zg^2]q_{(3)}^{SB}(g) + a_2\left[q_{(3)}^{SB}(g)\right]^2/2 - t_{(3)}^{SB}(g) \tag{30}$$

As Eq (29) shows, manufacturers, except those of type $g$, can obtain a strictly positive and incremental information rent $2\int_{\underline{g}}^{g}(a_3 - 0.5z + \mu v)\tau q_{(3)}^{SB}(\tau)d\tau$. Similar to $\alpha$, in practice, $\mu$ depends on the bargaining powers of the government and the manufacturer.

**Proposition 6.** Given linear flexible-proportional benefit-sharing contract $\left\{t_{(3)}^{SB}(g), q_{(3)}^{SB}(g)\right\}$,

1. $\dot{\pi}_{M,(3)}^{SB}(g) \geq 0$, if $\mu \geq (0.5z - a_3)/v$;

2. $\dot{\pi}_{M,(3)}^{SB}(g) < 0$, if $\mu < (0.5z - a_3)/v$.

When $\mu \in [(0.5z - a_3)/v, 1/\bar{g}]$, the linear flexible-proportional benefit-sharing contract $\left\{t_{(3)}^{SB}(g), q_{(3)}^{SB}(g)\right\}$ can distinguish different types of manufacturers, and the manufacturer's profit increases as $g$ increases. Unlike the fixed-proportional benefit-sharing contract, the range $\mu$ represented by $\mu \geq (0.5z - a_3)/v$ is unrelated to $g$, and the upper threshold $\mu$ is higher than that of $\alpha \in [(z - 2a_3)\bar{g}/v, 1]$. Also, the value range of the proportion coefficient $\mu$ is usually far more expansive than that of $\alpha$. Therefore, under such conditions, the flexible contract performs better when compared to the fixed contract.

## 6. Coordination contract based on Nash bargaining model

Asymmetric information may distort resource allocation so that only a second-best contract can be attained between the principal and the agent. However, the second-best contract may not allow for the optimization of the total profit for both the principal and the agent. Therefore, in this section, a Nash bargaining model is applied to achieve Pareto improvement of both agent and principal's profits. Considering the incentive mechanisms in Section 5, which help reveal the true green-degree, we construct a coordination contract to investigate the strategies for the manufacturer and the government based on the Nash bargaining model.

### 6.1. Optimal incentive contract under symmetric information

Under symmetric information, the government has access to the manufacturer's private green-degree information. So, in this section, we assume that $g$ is observable by the government. The government and the manufacturer work together to determine the optimal output of green products and the optimal transfer payment. The proof of the optimal incentive contract $\{q^*(g), t^*(g)\}$ under symmetric information is provided in S2 Appendix.

$$q^*(g) = [a_1 - 2\underline{c} + vg + (a_3 - z)g^2]/a_2 \tag{31}$$

$$t^*(g) = \underline{\pi}_M - [a_1 - \underline{c} + (a_3 - 0.5z)g^2 - a_2 q^*(g)]q^*(g) \tag{32}$$

Where superscript * denotes the optimal result under symmetric information.

With symmetric information, the government can drive manufacturer's profit $\pi_M^*$ to $\underline{\pi}_M$, and we have

$$U_G^* = -\underline{\pi}_M + [a_1 - 2\underline{c} + vg + (a_3 - z)g^2 - 0.5a_2 q^*(g)]q^*(g) \tag{33}$$

$$\pi_T^* = [a_1 - 2\underline{c} + vg + (a_3 - z)g^2 - 0.5a_2 q^*(g)]q^*(g) \tag{34}$$

**Proposition 7.** The optimal allocation of resources and maximization of total profits can be realized under symmetric information. However, in this scenario, the government monopolizes the benefits. The manufacturer, as an agent, receives a transfer payment that only covers

reservation profit and production cost, without any additional benefits. Thus, information disclosure is unfavorable to the manufacturer.

## 6.2. Non-linear coordination contract based on Nash bargaining model

Under asymmetric information, the manufacturer obtains extra benefits in the form of information rent, but it can hinder the optimization of resource allocation. We modify the flexible contract introduced in Section 5.2 using the Nash bargaining model for achieving Pareto improvement for both principal and agent's profits.

Nash [72] proposed a multi-player bargaining model which Harsanyi and Mariotti [24, 73] extended to an asymmetric Nash bargaining model, expressed as:

$$(\text{P-N}) \ (u_1(x^*), u_2(x^*), \cdots, u_n(x^*)) = \arg\max \prod_{i=1}^{n} (u_i(x) - \underline{u}_i)^{\gamma_i}$$

$$s.t \ (u_1(x^*), u_2(x^*), \cdots, u_n(x^*)) \geq (\underline{u}_1, \underline{u}_2, \cdots, \underline{u}_n); \ (u_1(x^*), u_2(x^*), \cdots, u_n(x^*)) \in S \quad (35)$$

herein $u_i(x)$ indicates the utility function of decision-maker $i$, $[i = 1, \cdots, n]$. $\underline{u}_i$ is the utility at the start of the negotiation, $S$ is the negotiation range, and $\gamma_i$ is the bargaining power with $\sum_{i=1}^{n} \gamma_i = 1$. We assume that both the manufacturers and the government are risk-neutral, and their negotiation starts from $\pi_{M,(3)}^{SB}(g)$ and $U_{G,(3)}^{SB}(g)$, with bargaining powers of $\gamma$ and $1 - \gamma$, respectively. When comparing $q^*(g)$ in the optimal incentive contract with $q_{(3)}^{SB}(g)$ in the flexible contract, we see that only the $\bar{g}$-type manufacturer's output is not distorted and satisfies $q^{SB}(\bar{g}) = q^*(\bar{g})$. Other types of manufacturers' output are $q^{SB}(g) < q^*(g)$. To achieve optimization of overall profit in the Nash bargaining model, the output of the $g$-type manufacturer is defined as the optimal $q^*(g)$, and the decision variable is the transfer payment $t(g)$. Based on the Nash bargaining model, the government's utility and the manufacturer's profit can be expressed as

$$U_G = (vg - \underline{c} - 0.5zg^2)q^*(g) + a_2[q^*(g)]^2/2 - t(g) \quad (36)$$

$$\pi_M(g) = [a_1 - \underline{c} + (a_3 - 0.5z)g^2 - a_2q^*(g)]q^*(g) + t(g) \quad (37)$$

According to the asymmetric Nash bargaining model, the objective function of the non-linear coordination contract is

$$\underset{\{t(g)\}}{Max} \ U[t(g)] = \left[ U_G(g) - U_{G,(3)}^{SB}(g) \right]^{1-\gamma} \left[ \pi_M(g) - \pi_{M,(3)}^{SB}(g) \right]^{\gamma} \quad (38)$$

So, we have transfer payment as follows,

$$t^{NB}(g) = \gamma \left[ (vg - \underline{c} - 0.5zg^2)q^*(g) + a_2[q^*(g)]^2/2 - U_{G,(3)}^{SB}(g) \right] - (1$$
$$- \gamma)\left\{ [a_1 - \underline{c} + (a_3 - 0.5z)g^2 - a_2q^*(g)]q^*(g) - \pi_{M,(3)}^{SB}(g) \right\} \quad (39)$$

where superscript $NB$ denotes the solution of the Nash bargaining model. Benefits of the government and manufacturer can be calculated (The proof is shown in S3 Appendix) and

represented by:

$$U_G^{NB}(g) = (1 - \gamma)\Big\{[a_1 - 2\underline{c} + vg + (a_3 - z)g^2 - 0.5a_2q^*(g)]q^*(g) - \pi_{M,(3)}^{SB}(g)\Big\} + \gamma U_{G,(3)}^{SB}(g) \quad (40)$$

$$\pi_M^{NB}(g) = \gamma\Big\{[a_1 - 2\underline{c} + vg + (a_3 - z)g^2 - 0.5a_2q^*(g)]q^*(g) - U_{G,(3)}^{SB}(g)\Big\} + (1 - \gamma)\pi_{M,(3)}^{SB}(g) \quad (41)$$

herein $\pi_T^{NB}(g) = \pi_T^*(g)$.

**Proposition 8**. If the bargaining power of principal and agent is fixed, a Nash negotiation model can facilitate the improvement of the linear benefit-sharing contract. This non-linear contract $\{t^{NB}(g), q^*(g)\}$ optimizes resource allocation and achieves Pareto improvement for both parties. It also ensures a monotonous risk rate for government and manufacturers and provides incentives for higher green-degree manufacturers.

The negotiation sequence between the government and manufacturer is as follows. Firstly, considering their respective bargaining powers, both parties determine the coefficient of flexible-proportional benefit-sharing coefficient $\mu$. Secondly, the government proposes the contract $\Big\{t_{(3)}^{SB}(g), q_{(3)}^{SB}(g)\Big\}$. After that, the manufacturer chooses a contract in line with its green-degree. It is through such endeavors that the contract is optimized based on bargaining power factor $\gamma$, and ultimately achieves a favorable contract denoted as $\{t^{NB}(g), q^*(g)\}$.

## 7. Numerical analysis

To better understand the relationship between the benefit-sharing ratio $\alpha$, the flexible coefficient of benefit-sharing $\mu$, the green-degree $g$, and the benefits of manufacturer and government, simulations results are presented in this section. The two linear proportional contracts are compared while considering similar sharing proportions. Since green products are still emerging in developing regions, the number of $g$-type manufacturers may decrease as green-degree $g$ increases ($\partial f(g)/\partial g < 0$). Specifically, we consider $f(g) = -g + 1.5$ and $F(g) = -0.5g^2 + 1.5g$. It should be noted that $g \in [0,1]$. Other parameters are set as $a = 20$, $b = 2$, $h = 1$, $\underline{c} = 1$, $z = 3$, $v = 3$, so $\alpha \in [(z - 2a_3)\bar{g}/v, 1] = [2/3, 1]$ and $\mu \in [(0.5z - a_3)/v, 1/\bar{g}] = [1/3, 1]$. The value range of the proportion coefficient $\mu$ is as twice as that of $\alpha$.

### 7.1. Relationship between g and profit/utility in linear fixed-proportional benefit-sharing model

Fig 1(a) and 1(b) illustrate the impact of green-degree and the fixed benefit-sharing ratio on manufacturer profit and government utility. The relationship between g, $\alpha$, and total profit is presented in Fig 1(c).

As is shown in Fig 1(a), The manufacturer's profit function is concave in the green-degree, indicating that manufacturers in the early stages of green product development are easily motivated to enhance the green-degree. According to Proposition 3, the order quantity of green products decreases when a larger fraction of benefits is shared with the manufacturer. However, the profit of the manufacturer is positively correlated with $\alpha$, implying that the manufacturer is incentivized to produce greener products due to higher financial transfer payments by the government. When $g$ reaches a high level ($g = 0.9$), and the benefit-sharing proportion is high ($\alpha = 0.9$), the manufacturer's profit has a high growth rate with $g$ and $\alpha$. So, in mature green markets, a high benefit-sharing proportion enables the manufacturer to enhance product greenness. For the government, as shown in Fig 1(b), the expected benefit function is concave in the green-degree. When the green-degree is low (e.g., $g \in [0,0.5]$) with $\alpha = 0.7$, the government's benefit (ranging from 62.04 to 66.46) increases with $g$. For a low green-degree

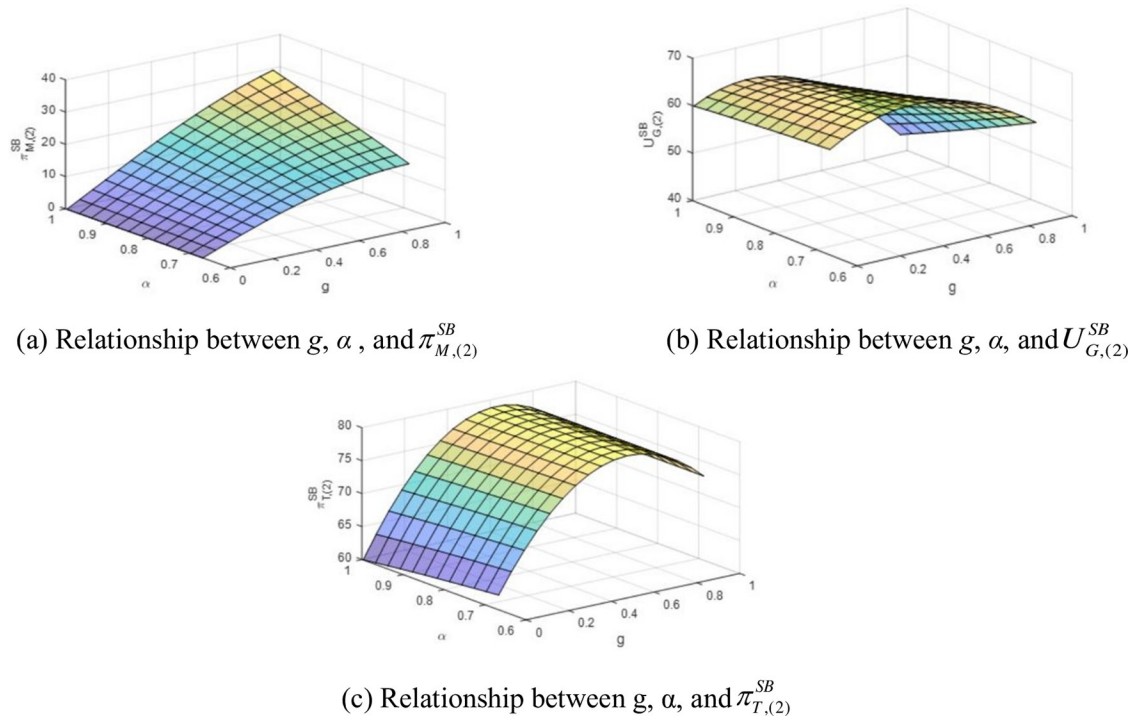

(a) Relationship between $g$, $\alpha$, and $\pi_{M,(2)}^{SB}$

(b) Relationship between $g$, $\alpha$, and $U_{G,(2)}^{SB}$

(c) Relationship between g, α, and $\pi_{T,(2)}^{SB}$

**Fig 1. Relationship between $g$, $\alpha$, and contractors' profits under linear fixed-proportional benefit-sharing contract.** (a) Relationship between $g$, $\alpha$, and $\pi_{M,(2)}^{SB}$, (b) Relationship between $g$, $\alpha$, and $U_{G,(2)}^{SB}$, (c) Relationship between $g$, $\alpha$, and $\pi_{T,(2)}^{SB}$.

manufacturer, the government's benefit changes marginally with different benefit-sharing proportions. However, the benefit declines with higher $\alpha$ if the manufacturers have products with high greenness. Fig 1(c) depicts the trend of total profit, which grows steadily with $g$ for the initially developing green product market and remains unchanged with $\alpha$.

**Observation 1**. In the fixed-proportional benefit-sharing contract $\left\{t_{(2)}^{SB}(g), q_{(2)}^{SB}(g)\right\}$, when the range of $\alpha$ is $[(z-2a_3)\bar{g}/v, 1]$, the manufacturer's profit increases with an increase in g and $\alpha$. The government's benefit and the total joint benefit of the manufacturer and government also increase with $g$ in the initial stages of green product development. The total collective benefit, however, is unrelated to $\alpha$. Therefore, the government has incentives to design and implement such contracts. Besides, the manufacturer's profit follows a monotonous risk rate with respect to both g and $\alpha$.

## 7.2. Relationship between g and profit/utility in linear flexible-proportional benefit-sharing contract

The parameters used in this section are the same as those in Section 7.1. The investigated range of proportion coefficient $\mu$ is $[(0.5z-a_3)/v, 1/\bar{g}] = [1/3, 1]$, which is twice that of $\alpha$. The relationships between proportion coefficient $\mu$, green-degree $g$, manufacturer's profit $\pi_{M,(3)}^{SB}$, government's utility $U_{G,(3)}^{SB}$, and total profit $\pi_{T,(3)}^{SB}$ are presented in Fig 2(a)–2(c).

Fig 2(a) shows that the manufacturer's profit is increasing and convex in $g$, and it increases at an increasing rate with $g$. The manufacturer's profit also increases with an increase in $\mu$, though the order quantity of green products reduces (according to Proposition 5). Comparing Figs 1(a) and 2(a), we notice that the growth rate of manufacturer profit is higher in the flexible-proportion contract compared to the fixed-proportion contract. This suggests that higher

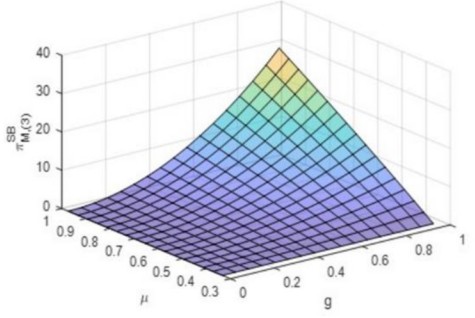

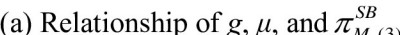

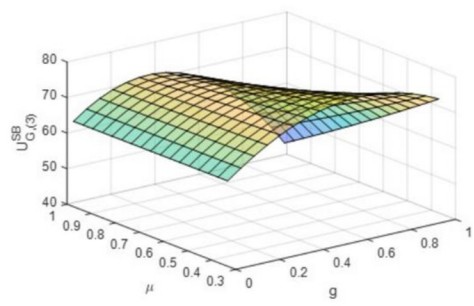

(a) Relationship of $g$, $\mu$, and $\pi_{M,(3)}^{SB}$    (b) Relationship of $g$, $\mu$, and $U_{G,(3)}^{SB}$

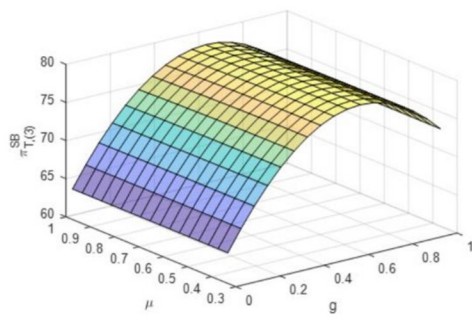

(c) Relationship between $g$, $\mu$, and $\pi_{T,(3)}^{SB}$

**Fig 2. Relationship between $g$, $\mu$, and contractors' profits under linear flexible-proportional benefit-sharing contract.** (a)
Relationship of $g$, $\mu$, and $\pi_{M,(3)}^{SB}$, (b) Relationship of $g$, $\mu$, and $U_{G,(3)}^{SB}$, (c) Relationship between $g$, $\mu$, and $\pi_{T,(3)}^{SB}$.

green-degree manufacturers may prefer a flexible-proportion benefit-sharing contract. As
shown in Fig 2(b), the government's utility increases with $g$ when $\mu$ is small. However, when $\mu$
is high, the manufacturers gain at the government's expense, whose benefit is concave in the
green-degree and experiences a steep decline when $g$ is high. The utility remains almost
unchanged with $\mu$ when $g$ is small, but it decreases when $g$ is high. The rate of decrease in the
government's share accelerates if $\mu$ rises because of rapidly increasing transfer payment to the
greener manufacturer. Manufacturers gain a higher profit share when both $g$ and $\mu$ are high.
The trend of total profit in Fig 2(c) is similar to that in Fig 1(c), where an increasing $g$ is benefi-
cial to the total profit in the early stages of green product development.

**Observation 2**. In the case of a linear flexible-proportional benefit-sharing contract
$\left\{ t_{(3)}^{SB}(g), q_{(3)}^{SB}(g) \right\}$, as $\mu = [(0.5z - a_3)/v, 1/\bar{g}]$ gets higher, the manufacturer's profit rises. The
government's utility remains almost unchanged with the change in $\mu$ when $g$ is small, but it
decreases if $g$ is at a high level. The total profit increases with $g$ when the manufacturer is not at
a high green level, and it remains the same under different benefit-sharing contracts.

### 7.3. Comparison of the two benefit-sharing contracts

We compare the fixed- and flexible-proportional contracts under similar scenarios when $\alpha$ is
equivalent to $\mu$. Let $\alpha = 0.8$ and $\mu = \alpha/\bar{g} = 0.8$. Other parameters are kept the same. Fig 3(a)

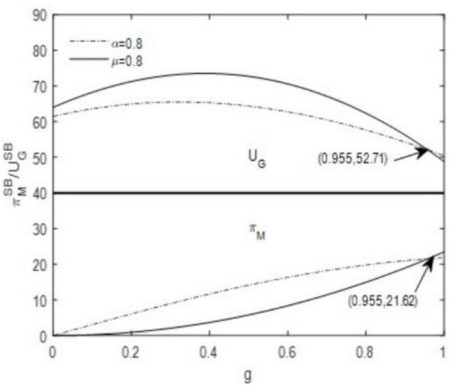 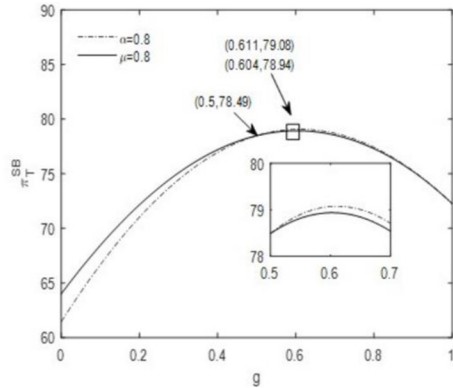

(a) Relationship of $\alpha$, $\mu$, $\pi_M^{SB}$ and $U_G^{SB}$     (b) Relationship of $\alpha$, $\mu$ and $\pi_T^{SB}$

**Fig 3. Comparison between fixed-proportional contract and flexible-proportional contract.** (a) Relationship of $\alpha$, $\mu$, $\pi_M^{SB}$ and $U_G^{SB}$, (b) Relationship of $\alpha$, $\mu$ and $\pi_T^{SB}$.

shows the interactions of $\alpha$ (fixed-proportion) and $\mu$ (flexible-proportion) with profits of the manufacturer and the government. For most values of $g$, the manufacturer's profit under $\alpha = 0.8$ is higher than that under $\mu = 0.8$. However, when $\mu = 0.8$, the manufacturer's profit is increasing and convex in $g$ and exceeds the profit with $\alpha = 0.8$ when $g$ reaches 0.955. This indicates that the manufacturer can be stimulated to achieve a higher green-degree under the flexible-proportional benefit-sharing contract. The government utility with $\mu = 0.8$ is greater than that with $\alpha = 0.8$; however, at high levels of $g$, the profit with $\alpha = 0.8$ becomes higher because more benefits are shared with the manufacturer in the flexible-proportional contract. The total profit in the fixed-proportional contract is lower at first; however, as $g$ gets higher ($g > 0.5$), a fixed-proportional contract leads to a slightly higher profit than the flexible-proportional contract. See Fig 3(b). The total profits under the fixed- and flexible-proportional contracts are highest when $g$ is 0.611 and 0.604, respectively.

When $\alpha$ is similar to $\mu$, the manufacturer's profit under a fixed-proportional benefit-sharing contract is generally greater than that under a flexible proportion contract. However, manufacturers are more encouraged to produce products of high green-degree in the flexible proportion contract. The utility of government under a flexible-proportion contract is higher within a specific range of $g$, but it is exceeded by a fixed-proportion contract when $g$ reaches a certain threshold. Both contracts can achieve almost identical maximum total profits. In general, governments in developing countries may choose a flexible-proportion contract because it leads to higher profit (for both the government alone and the two parties together) in the initial development stages of green products and provides more accessible incentives for manufacturers to improve greenness.

## 8. An application to inkjet printer remanufacturing

In this section, we present an example of inkjet printers to outline the practical implications of our model and findings. The case study helps to demonstrate the screening function for different green-degree products and the motivation for improving the environmental performance of these products. We also expound the results if the government agency negotiates with the manufacturer based on Nash bargaining.

## 8.1. Case background

Global printer demand has been increasing, with China alone producing ~5 million inkjet printers in 2018 [74]. In China, a prominent class of green products that government agencies procure is office equipment, including inkjet printers widely used in government agencies, public institutions, academic institutions, and corporations. The usable life of a printer is relatively short, and they are frequently replaced. Moreover, toner and ink cartridges are potentially hazardous to the environment. If not treated, recycled, or appropriately remanufactured, waste from printers that contain heavy metal contaminants and organic pollutants can aggravate pollution to groundwater, soil, and atmosphere [75, 76]. This poses a challenge and opportunity for the government to control the environmental impact of printers.

Many firms, such as Hewlett-Packard (HP) and Yidasheng Co., Ltd., have applied remanufacturing technology to operationalize environmental protection and resource conservation objectives. HP reconditioned 30 million printers and accessories from 2003 to 2012 and remanufactured 3.9 million hardware products that weighed more than 2,600 tons in 2012 [77]. Yidasheng Co., Ltd. was set up in Shanghai, China in 2001. The company specializes in the manufacturing and processing of printer consumables, including ribbons, ink cartridges, and toner cartridges. Additionally, they offer a wide range of computer products and accessories, along with office supplies. With a focus on remanufacturing toner and ink cartridges, the company engages in the exchange of used cartridges for remanufactured ones. The collected cartridges are processed by cleaning, drying, vacuum ink refilling, printing test, and appearance treatment to restore remanufactured products to guaranteed high quality [78].

A practical case of remanufactured printer procurement is introduced to illustrate implementing the contracts proposed in this paper. Let us assume a government agency in China needs 300 inkjet printers. Through a bidding process that factors in manufacturers' reputation, effort on environment protection, end-of-life product recycling, quality of remanufactured components, production scale, and financial performance, the agency decides to procure inkjet printers from a local Company X. Following *the Guide of recycle development* [15], government agencies are required to implement a program of preferred procurement for remanufactured products. We assessed the green-degree of printer manufacturers in China through inputs from interviews and our interactions with company experts $g \in [0.25, 0.5]$. The probability density function and cumulative probability density function of the remanufactured printer's green-degree are set as $f(g) = -8g + 7$ and $F(g) = -4g^2 + 7g - 1.5$, respectively.

As our results have revealed, government bodies in developing regions prefer a flexible-proportion benefit-sharing contract with printer manufacturing companies because China's remanufacturing market is still in its initial stage of development. Therefore, the flexible-proportion benefit-sharing contract is applied to the case. Based on interviews and interactions with companies in China as well as consideration on production scale and cost of remanufactured printers, we set the parameters as $a = 320$, $b = 4$, $h = 2$, $v = 78$, $z = 8$, $\underline{c} = 21$, and $\mu = 0.3$.

## 8.2. Implementation of a linear flexible-proportional benefit-sharing contract

In this section, we analyze the interactions of a government agency with a printer manufacturer (Company X). We assume that the true green degree of company X is 0.35; however, this information is private and is unobservable by the government agency. To cover the true green degree, we present solutions on the range of its $g$ from 0.25 to 0.5. This allows us to examine whether the flexible-proportional benefit-sharing contract can screen different green degrees and whether a manufacturer with a higher green-degree can receive higher profit. The

**Table 1. Procurement quantity, transfer payment, government's benefit, and manufacturer's profit with different green-degrees under flexible-proportional benefit-sharing contracts.**

| $g$ | 0.25 | 0.3 | 0.35 | 0.4 | 0.45 | 0.5 |
|---|---|---|---|---|---|---|
| $q_{(3)}^{SB}(g)$ | 220 | 235 | 250 | 266 | 283 | 289 |
| $t_{(3)}^{SB}(g)(\$)$ | -874 | 60 | 897 | 1,935 | 3,238 | 4,677 |
| $\pi_{M,(3)}^{SB}(\$)$ | 274 | 421 | 610 | 847 | 1,139 | 1,494 |
| $U_{G,(3)}^{SB}$ | 6,222 | 6,945 | 7,672 | 8,395 | 9,105 | 9,791 |

government's decision variables and benefits, along with Company X's profit under different green-degrees, are presented in Table 1.

As indicated in Table 1, the green-degree has a positive impact on procurement volume. Note that the procurement quantity of the producer (at $g = 0.35$) is 250, which falls short of the government's demand for 300 printers in all. As a result, the remaining 50 printers are supplemented by traditional printers. If Company X enhances its green-degree, it will get more transfer payment and profit. Even with the substantial transfer payment to Company X, the government obtains a high utility that grows with $g$ in an emerging market of green products. When $g = 0.25$, the transfer payment value is negative, indicating that due to the low level of greenness, the manufacturer is required to make additional payments to the government.

A Company X with a true green-degree $g = 0.35$ can get \$588, \$601, and \$608 from inflating the green-degree to $\tilde{g} = 0.4$, $\tilde{g} = 0.45$, and $\tilde{g} = 0.5$, respectively. Moreover, if the company underreports its green-degree to the government, with $\tilde{g} = 0.25$ and $\tilde{g} = 0.3$, their profits are only \$542 and \$571, respectively (see S4 Appendix). In contrast, if the company truthfully reveals its green degree, it earns a profit of \$610. Table 1 and S4 Appendix indicate that the manufacturer can obtain higher profits by truthfully revealing its green degree compared to reporting a false level of greenness. This clearly shows that the flexible-proportion benefit-sharing contract encourages manufacturers to disclose their true green degrees.

The government receives the environmental benefit, manufacturer's revenue, and CS, and shares a part of this utility with the manufacturer. The government's total utility increases with green-degree, partly because of the increasing CS and environmental benefits. For instance, when $g = 0.35$, the government's total utility is \$7,672, which includes the environmental benefit $vgq(g)$ of \$6,834 and a CS of \$7,833. Despite transferring a significant portion to the manufacturer (\$1,616 consisting of transfer payment of \$898 and shared environmental benefit of \$718), the government generates a high total utility because of the high values of CS and $vgq(g)$. This shows that the manufacturer's greening efforts lead to a higher utility for the government too.

In summary, as the case study demonstrates that a flexible benefit-sharing contract encourages the inkjet printer manufacturer to truthfully revealing the product's green degree. The manufacturer also has incentives to increase the product's green-degree, as higher green degree values lead to higher profits for the manufacturer.

## 8.3. Implementation of non-linear coordination contract based on Nash bargaining model

The government could screen printer manufacturers of different green-degrees by using a flexible-proportional benefit-sharing contract. However, to achieve Pareto improvement for both government and manufacturer's benefits, the government agency may negotiate with Company X based on the Nash bargaining model. Similar to the value of $\mu$, Company X's

**Table 2. Procurement quantity, transfer payment, government's benefit and manufacturer's profit with different green-degrees under the non-linear coordination contract.**

| $g$ | 0.25 | 0.3 | 0.35 | 0.4 | 0.45 | 0.5 |
|---|---|---|---|---|---|---|
| $q^*(g)$ | 228 | 243 | 258 | 272 | 286 | 288 |
| $t^{NB}(g)(\$)$ | -206 | 819 | 1,993 | 3,320 | 4,805 | 6,453 |
| $\pi_M^{NB}(\$)$ | 278 | 425 | 614 | 850 | 1,141 | 1,495 |
| $U_G^{NB}$ | 6,227 | 6,949 | 7,676 | 8,398 | 9,106 | 9,792 |

bargaining power also depends on the enterprise-scale, brand reputation, market competitiveness, etc. We present results using a value of $\gamma = 0.5$. When $g$ ranges from 0.25 to 0.5, the solutions under the coordination contract are indicated in Table 2.

The comparison between Tables 1 and 2 reveals several additional findings. With the adoption of a non-linear coordination contract based on the Nash bargaining model, the procurement volume, transfer payment, Company X's profit and the government agency's utility all increase with $g$. In particular, when the green degree reaches its highest level ($g = 0.5$), the procurement volume under the flexible-proportional benefit-sharing contract is equal to that under the coordination contract. Besides, both Company X's profit and the government agency's utility are higher under the non-linear coordination contract, indicating the total utility can be improved if they bargain over the linear incentive contract. In this case, when $g = 0.35$, eight additional remanufactured printers are procured under the non-linear coordination contract. Additionally, Company X's profit experiences a slight increase, while the government agency's utility grows from \$7,672 to \$7,676. Based on these findings, it is evident that the government agency, using the flexible-proportional benefit-sharing contract, can leverage the non-linear coordination contract to achieve mutually beneficial outcomes for all parties involved.

## 9. Conclusion

In developing countries like China, India, and Brazil, there has been a growing recognition of the importance of reducing environmental pollution, promoting sustainable development, and encouraging green production. In line with these goals, recent government initiatives have aimed to create incentives for manufacturers to improve their green performance. This paper specifically focuses on exploring the incentive mechanisms between the government and manufacturers in order to screen manufacturers based on their green-degrees and encourage the enhancement of green-degree in their products.

Combining game theory with the principal-agent model, we designed and analyzed the incentive contracts under governmental green procurement initiatives. Our primary results are as follows:

Under a lump-sum transfer contract $\left\{ t_{(1)}^{SB}(g), q_{(1)}^{SB}(g) \right\}$, if $a_3 \geq 0.5z$, the government can identify different types of manufacturers and effectively motivate them to improve green-degree. However, the condition $a_3 \geq 0.5z$ may not be satisfied in evolving markets as green products are costly invested in their initial development stages, and consumer awareness about such products is low.

The linear fixed-proportional benefit-sharing contract $\left\{ t_{(2)}^{SB}(g), q_{(2)}^{SB}(g) \right\}$ and flexible-proportional benefit-sharing contract $\left\{ t_{(3)}^{SB}(g), q_{(3)}^{SB}(g) \right\}$ both incentivize manufacturers to reveal their green-degree and realize higher profits in premise of $\alpha \in [(z - 2a_3)\bar{g}/v, 1]$ and

$\mu \in [(0.5z - a_3)/v, 1/\bar{g}]$, respectively. We note that: (1) the lower threshold of $\alpha$ is constrained by the highest level of greenness but that of $\mu$ is unrelated to the green-degree; (2) the upper threshold of $\mu$ is higher than that of $\alpha$ since the value of $g$ is less than 1. Note that $\mu$ is unconstrained on the lower bound of green-degree $g$; however, it is limited by the upper bound $\bar{g}$. This means that a flexible contract could be easier to implement by the government in a developing country.

Compared with the fixed-proportion contract, the government may prefer a flexible-proportion contract when e bargaining power between the manufacturer and the government is fixed. A manufacturer with low green-degree products may embrace a fixed-proportion contract, while high green-degree manufacturers are better off with a flexible-proportion contract. In general, a small difference of the total profit exists for varying values of $g$ under the two contracts. When $g$ is at a low level, total profit under a flexible contract is a little higher than that under a fixed contract; otherwise, the fixed contract leads to higher profits.

Although information disclosure facilitates the optimal allocation of resources, such disclosure may be disadvantageous to the manufacturers. Based on the Nash negotiation model, we transformed the flexible contract into a non-linear contract $\{t^{NB}(g), q^*(g)\}$ to achieve Pareto improvement for manufacturers' profit and the government's benefit.

A lump-sum transfer contract may be suitable for the countries where consumers have a high awareness of green consumption and the green products market is mature. As for the developing economies such as China, linear fixed- and flexible- proportional benefit-sharing contracts should be preferred, both of which are effective in green-degree screening and improvement. From the perspective of benefits, the flexible-proportional contract outweighs the fixed contract and is more likely to be accepted by the government when the greenness level is at a lower point.

This study is subject to several limitations that present future research opportunities. A single manufacturer is considered in this paper, while the government could purchase the same type of products from multiple manufacturers. One could extend the model to incorporate manufacturers competition by including multiple manufacturers producing the same component under a Stackelberg game model. In this context, how should the government design incentive contract to screen and improve the green-degree and ensure benefits? What is the interaction between competition and benefits of the government and manufacturers? Moreover, our model is based on a single-period game without considering green design or green advertising. Generally, the contractual relationship and fulfillment of green production may last for a long time, and both the governments and manufacturers could dynamically adjust their decisions. It would be interesting to consider green activities such as green design and green advertising to study the issue of green-degree over multiple periods by constructing a differential game model. In addition, the manufacturer in our model is assumed to be risk neutral. Future research can explore the effect of risk preference or risk aversion on manufacturers' green production initiatives.

## Supporting information

**S1 Table. Summary of notations.**
(DOCX)

**S1 Fig. Consumer surplus (CS).**
(DOCX)

**S1 Appendix. Equivalent conversion of programming problem (P1).**
(DOCX)

**S2 Appendix. The optimal incentive contract under symmetric information.**
(DOCX)

**S3 Appendix. Solution of transfer payment in the Nash bargaining model.**
(DOCX)

**S4 Appendix. Explanation for revealing the true green-degree for Company X.**
(DOCX)

## Author Contributions

**Conceptualization:** Jiayang Xu, Jian Cao.

**Formal analysis:** Jiayang Xu, Sisi Wu.

**Funding acquisition:** Jiayang Xu, Jian Cao.

**Investigation:** Jiayang Xu, Sanjay Kumar.

**Methodology:** Jiayang Xu, Sisi Wu.

**Validation:** Sanjay Kumar.

**Visualization:** Jiayang Xu.

**Writing – original draft:** Jiayang Xu, Jian Cao.

**Writing – review & editing:** Sanjay Kumar, Sisi Wu.

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
