## [Decision Letter · Decision Letter 0]

24 May 2023

PONE-D-23-12007Optimal government and manufacturer incentive contracts for green production with asymmetric informationPLOS ONE

Dear Dr. Xu,

Thank you for submitting your manuscript to PLOS ONE. After careful consideration, we feel that it has merit but does not fully meet PLOS ONE’s publication criteria as it currently stands. Therefore, we invite you to submit a revised version of the manuscript that addresses the points raised during the review process.

We look forward to receiving your revised manuscript.

Kind regards,

Syed Abdul Rehman Khan, PhD

Academic Editor

PLOS ONE

Journal Requirements:

Reviewers' comments:

Reviewer's Responses to Questions

**Comments to the Author**

1. Is the manuscript technically sound, and do the data support the conclusions?

Reviewer #1: Partly

Reviewer #2: Yes

2. Has the statistical analysis been performed appropriately and rigorously? 

Reviewer #1: Yes

Reviewer #2: Yes

3. Have the authors made all data underlying the findings in their manuscript fully available?

Reviewer #1: Yes

Reviewer #2: Yes

4. Is the manuscript presented in an intelligible fashion and written in standard English?

Reviewer #1: No

Reviewer #2: Yes

5. Review Comments to the Author

Reviewer #1: comment 1: the author briefly mentions the barriers faced by manufacturers in producing green products, but does not delve into the details of these barriers or provide any evidence to support this claim. This section could benefit from more in-depth analysis and supporting evidence to justify the need for government intervention in promoting the production and use of green products.

Additionally, while the author mentions the use of mechanism design and principal-agent theory in contracting with manufacturers, it would be helpful to provide a brief overview of these theories for readers who may not be familiar with them.

Comment 2 : the review could be improved in a few ways. Firstly, the review could benefit from a more structured approach, such as breaking down each stream of literature into sub-topics or themes, to make it easier for readers to navigate and understand the literature. Secondly, the review could be more critical of the literature it presents. For example, it could identify any limitations or gaps in the literature, or discuss any contradictory findings across different studies. Finally, the review could be more concise by focusing on the most relevant and important studies in each stream of literature, rather than providing a long list of studies.

Comment 3: the model framework provides a clear and detailed analysis of the incentives for manufacturers to produce green products, with a specific focus on remanufactured products. However, the review could be improved by providing a more detailed explanation of the green-degree metric.

Comment 4: the use of a Nash bargaining model to achieve a Pareto improvement in the principal and agent's profits is an appropriate approach, I would suggest that the passage could be improved by providing a more detailed explanation of how the model works and how it relates to the research question.

Comment 5: In page 31 about Case background the paragraph then jumps to discussing Yidasheng, a company that trades in older cartridges for remanufactured ones, without much explanation or context. Furthermore, the paragraph ends with a statement about the government's preferred contract with printer manufacturers, but there is no explanation or discussion of what this contract entails or how it relates to the rest of the paragraph.

Comment 6: some of the findings presented in the conclusion seem contradictory or unclear. For instance, the authors state that a flexible-proportional contract is more effective than a fixed-proportion contract in screening and improving green-degree. Yet, they also mention that the total profit of the government and manufacturer under the two contracts is similar. It would be helpful if the authors could provide a more detailed explanation of these seemingly conflicting results.

the authors acknowledge several limitations of their study, which is commendable. However, they do not fully explore the implications of these limitations for the validity and generalizability of their findings. For example, they mention that their model only considers a single manufacturer, but they do not discuss how this might affect the reliability of their results. It would be beneficial if the authors could elaborate on these limitations and offer suggestions for how future research could address them.

Reviewer #2: 1. The abstract is well written, however it might use some editing to make it easier to understand. You must improve the first three lines' wording. These are crucial in helping the reader comprehend the study's primary issue or gap.

2. The introduction is well organized, but to make it clearer, include the most recent references from 2023, particularly the publications that were published in reputable journals like PLOS ONE. Further include following citations in your study to make your work more evident:

a. Khan, S. A. R., Ahmad, Z., Sheikh, A. A., & Yu, Z. (2023). Green technology adoption paving the way toward sustainable performance in circular economy: a case of Pakistani small and medium enterprises. International Journal of Innovation Science.

b. Chen, W., & Li, L. (2021). Incentive contracts for green building production with asymmetric information. International Journal of Production Research, 59(6), 1860-1874.

c. Rehman Khan, S. A., Ahmad, Z., Sheikh, A. A., & Yu, Z. (2022). Digital transformation, smart technologies, and eco-innovation are paving the way toward sustainable supply chain performance. Science Progress, 105(4), 00368504221145648.

d. Khan, S. A. R., Sheikh, A. A., Ashraf, M., & Yu, Z. (2022). Improving Consumer-Based Green Brand Equity: The Role of Healthy Green Practices, Green Brand Attachment, and Green Skepticism. Sustainability, 14(19), 11829.

e. Nielsen, I. E., Majumder, S., Sana, S. S., & Saha, S. (2019). Comparative analysis of government incentives and game structures on single and two-period green supply chain. Journal of Cleaner Production, 235, 1371-1398.

f. Liu, L., Wang, Z., & Zhang, Z. (2021). Matching-game approach for green technology investment strategies in a supply chain under environmental regulations. Sustainable Production and Consumption, 28, 371-390.

g. Both the theoretical development and the literary part are written effectively. When developing the hypotheses, keep theory in mind to give them a more solid, organized framework.

h. In addition, the analysis, discussion, and consequence sections could be improved while still adhering to the journal's rules. Last, I found certain structural and grammatical issues while reading the manuscript. Therefore, you are required to address these issues before the final submission.

6. PLOS authors have the option to publish the peer review history of their article (what does this mean?). If published, this will include your full peer review and any attached files.

Reviewer #1: **Yes: **Alhamzah F. Abbas

Reviewer #2: **Yes: **Adnan Ahmed Sheikh, Ph.D.

---

## [Author Response · Author response to Decision Letter 0]

8 Jul 2023

Dear editor and reviewers,

Thank you very much for your valuable suggestions. We appreciate the opportunity to address the reviewers’ comments and improve the quality of our manuscript. In accordance with your advice, we have revised the paper in detail to make it more refined, with responses to each providing comment in the meantime. Any revisions can be tracked in the file labeled ‘Revised Manuscript with Track Changes’ and are further illustrated sequentially in ‘Response to Reviewers’. The line numbers mentioned below refer to the line markings within the ‘Manuscript’ file. We address each of your comments and provide a detailed explanation of the changes we have implemented. 

Thanks again for your recommendations.

With best regards,

Jiayang Xu

Reviewer #1: 

Comment 1: the author briefly mentions the barriers faced by manufacturers in producing green products, but does not delve into the details of these barriers or provide any evidence to support this claim. This section could benefit from more in-depth analysis and supporting evidence to justify the need for government intervention in promoting the production and use of green products.

Additionally, while the author mentions the use of mechanism design and principal-agent theory in contracting with manufacturers, it would be helpful to provide a brief overview of these theories for readers who may not be familiar with them.

Response: Thank you for your constructive criticism regarding the section on barriers faced by manufacturers in producing green products. We appreciate your attention to detail and your suggestion for more in-depth analysis and supporting evidence. To address this concern, we revise Section 1 to include a detailed analysis of the barriers (Lines 38-48), such as high cost, lack of technology, and limited consumer perception and demand for green products. 

One of the significant barriers is the higher production costs associated with green products. Eco-friendly materials and technologies often come at a higher price compared to conventional alternatives. Manufacturers may face difficulties in implementing sustainable practices and transitioning to greener alternatives due to the initial investment required [9]. Additionally, manufacturers may also face technical difficulties in integrating new equipment or processes into their existing production systems when implementing green manufacturing processes[10]. Adapting to and optimizing these technologies requires expertise, research, and development, which can be time-consuming and costly. While there is a growing awareness and interest in green products, the demand for such products may still be limited compared to conventional alternatives. Consumers may prioritize factors such as price, convenience, and product performance over environmental considerations[11]. Manufacturers face the challenge of creating green products that meet consumer expectations in terms of cost, quality, and functionality to drive wider adoption. 

To justify the need of government intervention in promoting the production and use of green products, we discuss the positive impacts of government policies and incentives in Lines 49-60. 

The development of green product not only relies on the power of the market itself, but also needs government guidance. Governments are assuming an increasingly significant role in the quest for sustainability, seeking to minimize the social and environmental impacts caused by production[12]. Therefore, the government agencies have enacted laws that encourage procuring and using green products. For example, the European Union (EU) has released Energy-using Products (EuP, 2005/32/EC), requiring that manufacturers should adopt the eco-friendly green design of product [13]. The US Environmental Protection Agency has developed environmentally preferable purchasing guidelines that focus on reducing raw material consumption, producing using renewable materials, and recycling used products [14]. A Guide of Recycle Development by the State Council in China mandates the establishment of a government preferential procurement system for remanufactured goods [15].In general, environmental regulations effectively guide manufacturers towards green production transformation and pollution reduction though directly affecting production decision in firm level, such as resource reallocation, capital investment and innovation incentives.

In addition, we illustrate the significance of government’s decision on how to subsidize to green product in the context of asymmetric information by introducing typical cases of illegal cheating of financial subsidy for new energy vehicles in China (Lines 79-86). 

For example, in 2016, the ministry of finance of China announced a special inspection of subsidies for the promotion and application of new energy vehicles and publicly exposed five typical cases of illegal cheating of financial subsidy for new energy vehicles. Notably, the largest reported amount of fraudulent subsidies amounted to 520 million CNY (~73 million $). Given the asymmetry of green-degree information, the government faces challenges in determining the appropriate level of subsidies and how they should be allocated. Consequently, finding solutions to prevent fraudulent subsidy claims by manufacturers and ensuring the efficient and equitable utilization of government's green subsidies becomes an immediate priority.

The description of mechanism design and principal-agent theory is also revised and supplemented.

It can be find that an incentive mechanism may effectively avoid the waste of government subsidy expenditure and unreasonable use when the government cannot clearly know the green-degree information of enterprises. On the premise that the green-degree of products can be improved, this paper uses the principal-agent theory to study the design of incentive contracts between government agencies and manufacturers who have private information about their product's green-degree. In this mechanism, the government is the principal and the manufacturer is the agent. The government first designs a set of contract menus. After the manufacturer observes the contract, it chooses according to the contract requirements (Lines 87-93).

Comment 2 : the review could be improved in a few ways. Firstly, the review could benefit from a more structured approach, such as breaking down each stream of literature into sub-topics or themes, to make it easier for readers to navigate and understand the literature. Secondly, the review could be more critical of the literature it presents. For example, it could identify any limitations or gaps in the literature, or discuss any contradictory findings across different studies. Finally, the review could be more concise by focusing on the most relevant and important studies in each stream of literature, rather than providing a long list of studies.

Response: According to your recommendations, we have decided to reconstruct literature review by adopting a structured approach. We have critically evaluated the literature and provided concise summaries of the most relevant studies. The literature is categorized into three streams: incentive contracts under asymmetric information, game analysis on green product improvement, and incentives of government intervention. Each stream of review is illustrated from several themes. 

The first stream is divided into two themes: the effect of different information asymmetry on the decision-making of participants and contracts adopted under asymmetric information. The gaps of existing literature and our research are as follows: our research primarily concentrates on the information asymmetry concerning manufacturers’ green-degree and considers government transfer payment as a main decision -making variable. Besides, we primarily propose a flexible proportional profit-sharing contract in which the shared percentage is related to the green-degree of the products.

The themes of the second stream include green-degree evaluation and decision-making processes of green products based on the framework of game theory. Our research expands the previous works of game modelling by introducing incentive contracts based on principal-agent theory. Instead of monitoring and evaluating the green-degree of products, we utilize the principal-agent model to screen the asymmetric green-degree information.

We have replaced the last stream with incentives of government intervention. We have summarized literature related to government policy or decision-making on manufacturers’ green production. Studies on government green procurement have also been discussed in this section. We have found that few researches apply incentive contracts between the government and enterprises to solve the problem of subsidy and green production strategy under asymmetry information. Thus, in the context of government subsidy and green procurement, this paper investigates the transfer payment of government and green production strategy of manufacturer through the principal-agent contract.

Comment 3: the model framework provides a clear and detailed analysis of the incentives for manufacturers to produce green products, with a specific focus on remanufactured products. However, the review could be improved by providing a more detailed explanation of the green-degree metric.

Response: The literature about green-degree metric is added in Section 2.2. 

Green product development has become an important issue due to the direct impact of green products on the environment. Achieving green products requires a comprehensive examination of environmental issues throughout the entire life cycle, from design to the disposal phase of old products [46-48]. Thus, the evaluation of the green degree of products is crucial. Xu et al.[49] provided a quantitative evaluation method to measure the green degree of different products of the same use function with an indicator system established, including fundamental indicators, general indicators, and leading indicators. Wang et al.[50] introduced a comprehensive method that integrates Fuzzy Extent Analysis and Fuzzy TOPSIS for the assessment of environmental performance with respect to different product designs. As a kind of increasing popular green products, the manufactured products’ sustainability performance evaluation have been extensively studied. Golinska et al.[51] classified remanufacturing sustainability performance by taking energy consumption level, waste generation level, material recovery rate and generated emissions level as criteria. Xu [52] developed an assessment model of resource and environmental benefit for the remanufacturing of decommissioned construction machinery to analyze its energy, materials and carbon dioxide emissions. (Lines 147-159).

Comment 4: the use of a Nash bargaining model to achieve a Pareto improvement in the principal and agent's profits is an appropriate approach, I would suggest that the passage could be improved by providing a more detailed explanation of how the model works and how it relates to the research question.

Response: Here, we would like to explain why we construct Nash bargaining model. We have confirmed the effect of lump-sum transfer and benefit-sharing contracts on screening and incentivizing green production. However, optimization of the government and manufacturer’s profit can not be attained in the contracts because asymmetric information distorts resource allocation. Therefore, we first propose an incentive contract under symmetric information, which realizes the maximization of total profits but results in principal’ monopoly on benefits. Based on the flexible contract, we furtherly propose asymmetric Nash bargaining model assuming that the enterprise has ability to negotiate with the government. The result reveals the Pareto improvement of both parties and the overall benefit optimization of the supply chain in Nash bargaining model.

Actually, the maximum value of total profit obtained in incentive contract under symmetric information is to be compared to the value of total profit in asymmetric Nash bargaining model. We get the conclusion that a Pareto improvement can be achieved under asymmetric Nash bargaining model. In sum up, the non-linear coordination contract based on Nash bargaining solution is an improvement over the flexible-proportional contract. The total benefit of both the government and the manufacturer is higher under Nash bargaining model if the bargaining power of principal and agent is fixed. The relation between Nash bargaining model and research topic is supplemented. In the abstract, we elucidated the purpose of the Nash bargaining model and its relationship with the flexible contract (Lines18-19, Lines 23-24). In Section 8.2 and Section 8.3 , we illustrate the application of the Nash bargaining model in comparison to the flexible-proportional benefit-sharing contract.

The process of Nash bargaining model implementation is illustrated at the end of Section 6.

The negotiation sequence between the government and manufacturer is as follows. Firstly, considering their respective bargaining powers, both parties determine the coefficient of flexible-proportional benefit-sharing coefficient μ. Secondly, the government proposes the contract . After that, the manufacturer chooses a contract in line with its green-degree. It is through such endeavors that the contract is optimized based on bargaining power factor γ , and ultimately achieves a favorable contract denoted as .

Comment 5: In page 31 about Case background the paragraph then jumps to discussing Yidasheng, a company that trades in older cartridges for remanufactured ones, without much explanation or context. Furthermore, the paragraph ends with a statement about the government's preferred contract with printer manufacturers, but there is no explanation or discussion of what this contract entails or how it relates to the rest of the paragraph.

Response: We are very sorry for our negligence of missing the introduction of the company. We have provided additional details about the company’s core business and remanufacturing activities in Lines 673-679. We have used Hewlett-Packard and Yidasheng Co., Ltd as examples to illustrate that there are indeed enterprises currently engaged in remanufacturing printers. This demonstrates the rationale and significance of choosing remanufactured printers as the subject of our case study.

Since China’s remanufacturing market is still in its initial stage of development, based on the model and numerical analysis, we have concluded that flexible contracts are more suitable for developing countries. Therefore, we directly apply the flexible contract to this case, and confirm that the flexible contract can identify and improve green-degree in combination with actual cases. The reason for using flexible-proportion benefit-sharing contract in the case analysis has been briefly explained in Lines 690-693.

Comment 6: some of the findings presented in the conclusion seem contradictory or unclear. For instance, the authors state that a flexible-proportional contract is more effective than a fixed-proportion contract in screening and improving green-degree. Yet, they also mention that the total profit of the government and manufacturer under the two contracts is similar. It would be helpful if the authors could provide a more detailed explanation of these seemingly conflicting results.

the authors acknowledge several limitations of their study, which is commendable. However, they do not fully explore the implications of these limitations for the validity and generalizability of their findings. For example, they mention that their model only considers a single manufacturer, but they do not discuss how this might affect the reliability of their results. It would be beneficial if the authors could elaborate on these limitations and offer suggestions for how future research could address them.

Response: Thank you for your valuable suggestions. As shown in Figure 3(b), instead of being similar, there is a slight difference in the total profit under the two contracts for varying values of g. We have removed the previously incorrect expression. The revised conclusion is as follows:

In general, a small difference of the total profit exists for varying values of g under the two contracts. When g is at a low level, total profit under a flexible contract is a little higher than that under a fixed contract; otherwise, the fixed contract leads to higher profits (Lines779-781).

We acknowledge that we did not fully explore the implications of the limitations for the validity and generalizability of our findings and offer suggestions for future research. For the first limitation, we demonstrated that the competition among manufacturers would affect the government’s incentive contract design. The interaction of green-degree and benefits of the government and manufacturers may also be different when companies compete with each other. The existing model could be extended to incorporate manufacturers competition by including multiple manufacturers producing the same component under a Stackelberg game model. The second limitation is that our model is based on a single-period game without considering long-term dynamics. In this context, both the governments and manufacturers could dynamically adjust their decisions, so their decision-making may vary with time. We address this issue by developing a differential game model between the government and the manufacturer. Furthermore, we identified an additional limitation. The manufacturer in our model is assumed to be risk neutral. Future research can explore the effect of risk preference or risk aversion on manufacturers’ green production initiatives.

Reviewer #2 

Comment 1: The abstract is well written, however it might use some editing to make it easier to understand. You must improve wording. These are crucial in helping the reader comprehend the study's primary issue or gap.

Response: Thank you for your suggestions. The first three lines have been revised to provide a clear background of our research. Here’s a revised version of the abstract to improve clarity and help the reader understand the primary issue of the study:

Governments commonly utilize subsidy policy to incentivize manufacturers to produce green products, promoting sustainable development. However, in the presence of information asymmetry, some manufacturers may dishonestly misrepresent the green degree of their products to secure higher subsidies. This study examines different incentive contracts between the government and a green product manufacturer who keeps private information of a product's green-degree in a principal-agent model. Lump-sum transfer and fixed- and flexible-proportion benefit-sharing contracts are proposed to investigate screening and improving green-degree issues. To further enhance the flexible-proportion benefit-sharing contract, we construct a non-linear coordinated contract based on the Nash bargaining solution. The revelation principle and Nash bargaining are performed for comparison and analysis of the contracts. We find that the lump-sum contract reveals true green-degree information but fails to impel manufacturers to improve product’s green-degree in developing countries where green product development is in initial stages. In contrast, both fixed- and flexible- proportion benefit-sharing contracts are effective in reveling and enhancing green-degree. The non-linear coordination contract optimizes resource allocation and achieves Pareto improvement. An applied case study for inkjet printer operations and numerical experiments corroborate our model findings.

Comment 2: The introduction is well organized, but to make it clearer, include the most recent references from 2023, particularly the publications that were published in reputable journals like PLOS ONE. Further include following citations in your study to make your work more evident:

a. Khan, S. A. R., Ahmad, Z., Sheikh, A. A., & Yu, Z. (2023). Green technology adoption paving the way toward sustainable performance in circular economy: a case of Pakistani small and medium enterprises. International Journal of Innovation Science.

b. Chen, W., & Li, L. (2021). Incentive contracts for green building production with asymmetric information. International Journal of Production Research, 59(6), 1860-1874.

c. Rehman Khan, S. A., Ahmad, Z., Sheikh, A. A., & Yu, Z. (2022). Digital transformation, smart technologies, and eco-innovation are paving the way toward sustainable supply chain performance. Science Progress, 105(4), 00368504221145648.

d. Khan, S. A. R., Sheikh, A. A., Ashraf, M., & Yu, Z. (2022). Improving Consumer-Based Green Brand Equity: The Role of Healthy Green Practices, Green Brand Attachment, and Green Skepticism. Sustainability, 14(19), 11829.

e. Nielsen, I. E., Majumder, S., Sana, S. S., & Saha, S. (2019). Comparative analysis of government incentives and game structures on single and two-period green supply chain. Journal of Cleaner Production, 235, 1371-1398.

f. Liu, L., Wang, Z., & Zhang, Z. (2021). Matching-game approach for green technology investment strategies in a supply chain under environmental regulations. Sustainable Production and Consumption, 28, 371-390.

Response: Thank you for your suggestions and references that can make the introduction more comprehensive. We cited five of the papers you mentioned above in Line 38, Line 41 and Line 43 on the topic of environmental-related issues, technology and demand of green products, and cost of green transformation, respectively.

We have consulted several recent articles from PLOS ONE:

a. He L, Wang M. Environmental regulation and green innovation of polluting firms in China. PLoS ONE.2023; 18(3): e0281303.

b. Ma L, Li Z, Zheng D. Analysis of Chinese consumers’ willingness and behavioral change to purchase green agri-food product online. PLoS ONE.2022; 17(4): e0265887.

c. Lu L, Su X, Hu S, Luo X, Liao Z, Ren Y, et al. Green transition in manufacturing: Dynamics and simulation. 2023; PLoS ONE 18(1): e0280389.

d. Pedram A, Pedram P, Yusoff NB, Sorooshian S. Development of closed-loop supply chain network in terms of corporate social responsibility. PloS ONE. 2017;12(5): e0178723.

g. Both the theoretical development and the literary part are written effectively. When developing the hypotheses, keep theory in mind to give them a more solid, organized framework.

Response: We appreciate your suggestion to keep theory in mind while developing the hypotheses in order to provide a more solid and organized framework. We completely agree with this suggestion and have taken it into consideration. 

In section 4 and section 5, we propose a lump-sum transfer contract and two benefit-sharing contracts. After solving the contract problem, we have developed two propositions for each model. These two propositions respectively indicate the relationship between production quantity and mutual benefits with green degree, as well as the conditions for effective implementation of the contract.

Because optimization of the government and manufacturer’s profit can not be attained in the contracts because asymmetric information distorts resource allocation. Therefore, in section 6, we propose an optimal incentive contract under symmetric information and a non-linear coordination contract based on Nash bargaining model. The result reveals the Pareto improvement for both principal and agent’s profits. 

Then, we conduct numerical analysis to make readers better understand the proposed contracts. Relationship between green-degree and profit/utility in the benefit-sharing models are observed. By comparing the fixed- and flexible-proportional contracts under similar scenarios when α is equivalent to , we have found that governments in developing countries may choose a flexible-proportion contract because it leads to higher profit (for both, government alone and the two parties together) in the initial development stages of green products and more accessible incentives to greenness improvement for manufacturers.

Based on the conclusions drawn above, we apply the flexible-proportion contract and the non-linear coordination contract based on Nash bargaining model to an example of inkjet printers to outline the practical implications of our model and findings. In this case study, we further demonstrate that the Nash bargaining model represents an improvement over the flexible contract.

h. In addition, the analysis, discussion, and consequence sections could be improved while still adhering to the journal's rules. Last, I found certain structural and grammatical issues while reading the manuscript. Therefore, you are required to address these issues before the final submission.

Response: We appreciate your suggestions regarding the analysis, discussion, and consequence sections of our manuscript. We have carefully reviewed these sections and make the necessary improvements while ensuring compliance with the journal's guidelines and requirements.

We also appreciate you bringing the structural and grammatical issues to our attention. We have conducted a thorough review and made the necessary revisions to address these issues before the final submission.

We want to apologize for the bad reading experience caused by some language mistakes. We have checked and corrected English grammar and spelling errors thoroughly. One of authors, Sanjay Kumar, a native English speaker, has corrected the syntactic and grammatical errors in our paper. We have checked the manuscript’s structure, formatting, citations, references, figures, tables, and supplementary materials to align with the guidelines provided by PLOS ONE.

In conclusion, we would like to express our sincere appreciation for the reviewers’ time, effort, and expertise in reviewing our manuscript. Your constructive suggestions have undoubtedly strengthened the quality and impact of our work. We hope that the revisions we have made address all the concerns raised and meet your expectations.

Should you have any further questions or require additional clarification, please do not hesitate to reach out to us. We remain committed to the continual improvement of our research and appreciate any further guidance you may provide.

Thank you once again for your invaluable contribution to our manuscript.

Sincerely,

Jiayang Xu

---

## [Decision Letter · Decision Letter 1]

24 Jul 2023

Optimal government and manufacturer incentive contracts for green production with asymmetric information

PONE-D-23-12007R1

Dear Dr. Xu,

We’re pleased to inform you that your manuscript has been judged scientifically suitable for publication and will be formally accepted for publication once it meets all outstanding technical requirements.

Kind regards,

Syed Abdul Rehman Khan, PhD

Academic Editor

PLOS ONE

Reviewers' comments:

Reviewer's Responses to Questions

**Comments to the Author**

1. If the authors have adequately addressed your comments raised in a previous round of review and you feel that this manuscript is now acceptable for publication, you may indicate that here to bypass the “Comments to the Author” section, enter your conflict of interest statement in the “Confidential to Editor” section, and submit your "Accept" recommendation.

Reviewer #1: All comments have been addressed

Reviewer #2: All comments have been addressed

2. Is the manuscript technically sound, and do the data support the conclusions?

Reviewer #1: Yes

Reviewer #2: Yes

3. Has the statistical analysis been performed appropriately and rigorously? 

Reviewer #1: Yes

Reviewer #2: Yes

4. Have the authors made all data underlying the findings in their manuscript fully available?

Reviewer #1: Yes

Reviewer #2: Yes

5. Is the manuscript presented in an intelligible fashion and written in standard English?

Reviewer #1: Yes

Reviewer #2: Yes

6. Review Comments to the Author

Reviewer #1: The authors did all the necessary comments. and I check all the paper and I believe the paper now is ready for publication.

Reviewer #2: Author has addressed all the major concern related to the corrections advised earlier. I will further suggest to recheck the grammatical and structural issues before final submission of the article

7. PLOS authors have the option to publish the peer review history of their article (what does this mean?). If published, this will include your full peer review and any attached files.

Reviewer #1: **Yes: **Alhamzah F. Abbas

Reviewer #2: **Yes: **Dr. Adnan Ahmed Sheikh

---

## [Editor Report · Acceptance letter]

28 Jul 2023

PONE-D-23-12007R1 

Optimal government and manufacturer incentive contracts for green production with asymmetric information 

Dear Dr. Xu:

I'm pleased to inform you that your manuscript has been deemed suitable for publication in PLOS ONE. Congratulations! Your manuscript is now with our production department. 

Kind regards, 

on behalf of

Dr. Syed Abdul Rehman Khan 

Academic Editor

PLOS ONE